# Temporal Latent Bottleneck: Synthesis of Fast and Slow Processing Mechanisms in Sequence Learning

**Aniket Didolkar [1], Kshitij Gupta [1], Anirudh Goyal [1], Nitesh B. Gundavarapu [5]**
**Alex Lamb [2], Nan Rosemary Ke [3], Yoshua Bengio [1,4]**

## Abstract

Recurrent neural networks have a strong inductive bias towards learning temporally compressed representations, as the entire history of a sequence is represented by a single vector. By contrast, Transformers have little inductive bias towards learning temporally compressed representations, as they allow for attention over all previously computed elements in a sequence. Having a more compressed representation of a sequence may be beneficial for generalization, as a high-level representation may be more easily re-used and re-purposed and will contain fewer irrelevant details. At the same time, excessive compression of representations comes at the cost of expressiveness. We propose a solution which divides computation into two streams. A slow stream that is recurrent in nature aims to learn a specialized and compressed representation, by forcing chunks of $K$ time steps into a single representation which is divided into multiple vectors. At the same time, a fast stream is parameterized as a Transformer to process chunks consisting of $K$ time-steps conditioned on the information in the slow-stream. In the proposed approach we hope to gain the expressiveness of the Transformer, while encouraging better compression and structuring of representations in the slow stream. We show the benefits of the proposed method in terms of improved sample efficiency and generalization performance as compared to various competitive baselines for visual perception and sequential decision making tasks.

## 1 Introduction

The interplay between fast and slow mechanisms for information processing and perception has been studied in both cognitive science and machine learning [5, 35]. In the brain, short-term and long-term memory have developed in a specialized way. Short-term memory is allowed to change very quickly to react to immediate sensory inputs and perception. It also tends towards high capacity storage of all pieces of information which may be relevant for future reasoning [42, 3, 4]. By contrast, long-term memory changes slowly [45, 41], is highly selective and involves repeated consolidation. It contains a set of memories that summarize the entire past, only storing details about observations which are most relevant [28, 6].

Deep Learning has seen a variety of architectures for processing sequential data [36, 57, 18]. For example. recurrent neural networks compress information about a sequence into a single hidden state. Transformers get rid of the recurrent state by dynamically capturing information between positions using multi-head dot product attention [61]. Transformers have become the dominant architecture across a wide range of domains including vision [25], natural language [24, 54, 11, 69, 20, 55], and reinforcement learning [13, 40]. They have eclipsed recurrent neural networks [36, 57, 18] in almost all sequence processing domains due to their high representational capacity and scalability. Despite their wide applicability, it is well known that Transformers are very data

---

[01] Mila, University of Montreal, [2] Microsoft Research, New York, NY, [3] Google Deepmind, [4] CIFAR Fellow, [5] Google Research, Corresponding authors: `adidolkar123@gmail.com`

36th Conference on Neural Information Processing Systems (NeurIPS 2022).

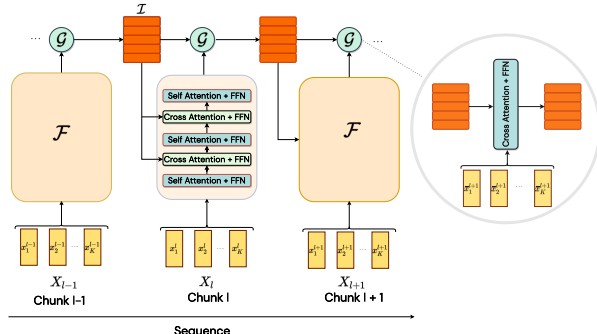

Figure 1: **Perceptual module + Temporal Latent Bottleneck Model**. $\mathcal{F}$ denotes the perceptual module or the fast stream which is a Transformer. $\mathcal{I}$ represents the temporal latent bottleneck state (consisting of a set of vectors) that are updated using a recurrent function denoted by $\mathcal{G}$. The given sequence is first divided into chunks of size $K$ and each chunk $X_l$ is processed by $\mathcal{F}$ which consists of interleaved SELF ATTENTION + FFN (denoted in blue) and CROSS ATTENTION + FFN (denoted in green) layers. The CROSS ATTENTION + FFN layers allow the representation of $\mathcal{F}$ to be conditioned on top-down information from $\mathcal{I}$. The representations of the temporal latent bottleneck state is updated using the outputs of $\mathcal{F}$ by a recurrent function $\mathcal{G}$, which consists of a CROSS ATTENTION + FFN layer as shown in the circle.

hungry and work well mainly at scale. This can be attributed to their inductive bias towards modeling all possible pairwise interactions in the sequence which results in no consolidation of information. This lack of selectivity in the attention mechanism also leads to a high computational complexity which scales quadratically with input size. Additionally, modeling all possible pairwise interactions maybe extremely wasteful and may result in capturing unnecessary information not useful for the downstream task [29, 39]. The goal of this work is to design an architecture for autoregressive modeling that has an inductive bias towards learning temporally compressed representation that retains the benefits of Transformers while preserving long-range interactions.

For learning temporally compressed representations, we start by dividing the computation of the Transformer into two streams of processing - a fast stream and a slow stream. Inspired by the idea of long-term and short-term memory, we want the fast stream to have a short-term memory with a high capacity that reacts quickly to sensory input. We refer to this fast stream as the perceptual module and implement it using a Transformer since they are known to have high representational capacity. On the other hand, we want the slow stream to have a long-term memory which updates at a slower rate and summarizes the most important information in the input sequence. We refer to this slow stream as the **Temporal Latent Bottleneck**.

Implementation-wise, we divide the input into fixed size chunks (Figure 1). The fast stream operates within each chunk while the slow stream consolidates and aggregates information across chunks updating itself once per chunk. This leads to *information asymmetry* between fast and slow stream as the fast stream contains fine-grained local information while the slow stream contains coarse-grained distant information. Such kind of information asymmetry has shown to improve generalization and adaptation performance of learned policies in the context of RL [30, 27]. The fast and slow streams interact with each other though bottleneck of attention. The division of computation into a fast and slow stream eliminates the need for capturing all possible pairwise interactions and thus introducing selectivity in the attention mechanism resulting in a much lower computational complexity which is not quadratic in the input size. We show that the limited capacity of the slow stream and consolidation of information by a recurrent neural network prevents the model from capturing unnecessary information not useful for the downstream task. We evaluate the proposed model in a number of domains showing that it consistently outperforms competent baselines showing improved generalization to scenarios not seen during training.

## 2 Methodology

We now present the proposed approach in detail. Our model jointly leverages the strengths of Transformers [61] and recurrent neural networks [18, 36].

### 2.1 Desiderata for Fast and Slow Streams of Processing

We give the detailed description of the proposed model in the next section. Here, we give an overview of our architecture and discuss some of its key properties. Given an input sequence, it

**Algorithm 1:** PyTorch-style pseudocode for proposed model

```
# C(query, key, value):  CROSS ATTENTION + FFN LAYER
# S(query, key, value):  SELF ATTENTION + FFN LAYER
# L: Num.  Layers
# R: Num.  C per S
# X: Input sequence of length T. shape:  [B x T x D]
# I:  The Temporal Bottleneck
# K: Chunk Size

X = torch.chunk(X, K, dim = 1) # List of length ⌊T/K⌋ with each element
 of size [B x K x D]

for X_c in X:
    for l in range(L):
        X_c = S^l(X_c,   X_c,   X_c)
        if l % R == 0:
            X_c = C^⌊L/l⌋(X_c,   I,   I)
    I = C(I,   X_c,   X_c)
```

is first divided into chunks of size $K$. Each chunk is processed by perceptual module represented by a Transformer (denoted as $\mathcal{F}$). While processing each chunk, $\mathcal{F}$ is also conditioned on information from the Temporal Latent Bottleneck module $\mathcal{G}$. The slow stream is a recurrent stream which has its own state consisting of a set of $N$ vectors (or slots) also called temporal latent bottleneck state denoted as $\mathcal{I}$ in Figure 1. In the following sections, we use the term *temporal latent bottleneck* to refer to the temporal latent bottleneck state $\mathcal{I}$. This state is updated once per chunk using information from the perceptual module through a cross attention mechanism.

The perceptual module operates within each chunk while the temporal latent bottleneck operates across chunks slowly updating itself after each chunk has been processed by the perceptual module. Thus, the only way the perceptual module gets information about inputs beyond its own chunk is through the temporal latent bottleneck. An added advantage of this is that the computational complexity of the attention mechanism in the proposed model is $\mathcal{O}(\frac{T}{K}(K^2 + KN))$ while that of a Transformer is $\mathcal{O}(T^2)$, where $T$ is the length of the sequence, $K$ is the chunk size, and $N$ is the number of temporal latent bottleneck state vectors. Since $K << T$ and $N << T$, we can see that $\frac{T}{K}(K^2 + KN) < T^2$. Therefore the proposed model has a much lower computational complexity compared to a Transformer. Furthermore, the capacity of the temporal latent bottleneck is limited and much smaller than that of the perceptual module. This encourages the temporal latent bottleneck to represent the most salient information about the past while the perceptual module represents only local information. This creates an *information asymmetry* between the two streams. This information asymmetry leads to the perceptual module having a fine grained view of the nearby inputs but a very coarse grained view of the distant past. This is very different from the usual self-attention which attends to all tokens in the sequence at the same level of granularity.

An advantage of having a compressed representation of the past is that it allows the model to forget irrelevant information. For example, if an agent is navigating in a large maze, it does not need to have fine grained knowledge of its actions from the distant past. In the case of a Transformer, it would attend to every step from the past (including steps from the distant past) which may be irrelevant in the present context thus wasting its capacity in modeling irrelevant details. Another important component of the proposed model is top-down attention which conveys contextual information from the high-level Temporal Latent Bottleneck module to the processing of low-level perceptual module. Past works [52, 26, 34, 23] have shown that top-down attention improves generalization and adaptation performance of the learned model. One difference between these works and the proposed model is that in their case the multiple streams operate at the same temporal granularity while in our case the streams operate at a different time scales (because of information asymmetry). Through our experiments, we show the advantage of the proposed architecture over these works. Next, we describe the detailed implementation of the proposed model.

## 2.2  Computational Steps

We denote the input $X$ as a sequence of $T$ tokens - $X = [x_0, x_1, x_2, \ldots, x_t]$. We chunk this input into chunks of size $K$ resulting in $\lfloor T/K \rfloor$ chunks. We refer to $l^{th}$ chunk as $X_l$. We represent the

state of the temporal latent bottleneck $\mathcal{I}$ (i.e. the slow stream) as a set of $M$ $d$-dimensional vectors. As mentioned previously, we denote the temporal latent bottleneck module as $\mathcal{G}$ and the perceptual module as $\mathcal{F}$. $\mathcal{G}$ updates the temporal latent bottleneck state while $\mathcal{F}$ processes chunks $X_l$ to form the latent representation $\bar{X}_l$ -

$$\text{Perceptual Module} \quad \bar{X}_l = \mathcal{F}(X_l, \mathcal{I}_l) \tag{1}$$

$$\text{Temporal Latent Bottleneck Module} \quad \mathcal{I}_{l+1} = \mathcal{G}(\mathcal{I}_l, \bar{X}_l) \tag{2}$$

**Preliminaries**. The central components of our model are the key value attention mechanism [7, 61] and the FFN module [61]. We use two forms of the attention mechanism -(1) Self Attention [61]: In this the query and key vectors refer to the same set of vectors; (2) Cross Attention [29, 39, 31]: In this the query and key vectors refer to seperate sets of vectors.

**Perceptual Module $\mathcal{F}$.** As mentioned previously, the perceptual module refers to the fast stream that acts directly on the input. The perceptual module operates on each chunk separately. Therefore, at any time the input to the perceptual module are the tokens corresponding to a particular chunk $X_l = [x_{l \times K}, x_{l \times K+1}, \ldots, x_{l \times K+K}]$. The perceptual module is a Transformer with self attention layers, cross attention layers, and FFNs. It has 2 kinds of layers - (1) SELF ATTENTION + FFN; (2) CROSS ATTENTION + FFN. The SELF ATTENTION + FFN layers process the input tokens and the CROSS ATTENTION + FFN layers integrate top-down information from the temporal latent bottleneck state $\mathcal{I}$ as follows -

$$X_l = \text{ATTENTION}(\text{LN}(X_l), \text{LN}(\mathcal{I}), \text{LN}(\mathcal{I})) + X_l$$
$$X_l = \text{FFN}(\text{LN}(X_l)) + X_l \tag{3}$$

We include one CROSS ATTENTION + FFN layer per $R$ SELF ATTENTION + FFN layers. The diagramatic representation of the perceptual module is presented in Figure 1 (in the processing of chunk $X_l$). In the figure, we set $R = 1$.

**Temporal Latent Bottleneck Module $\mathcal{G}$.** The temporal latent bottleneck (TLB) module represents the slow stream that operates on the temporal latent bottleneck state $\mathcal{I}$. $\mathcal{I}$ is updated using information from a particular chunk processed by the perceptual module. This update happens once for each chunk of the perceptual module resulting in $\lfloor T/K \rfloor$ updates for $\mathcal{I}$. Since the temporal latent bottleneck state $\mathcal{I}$ updates at a lower frequency than the perceptual module, it is expected to capture more stable and slowly changing features while the perceptual module captures faster changing features resulting in multiple scales of information representation. An update to the temporal latent bottleneck state $\mathcal{I}$ consists of a cross attention operation where the queries come from $\mathcal{I}$ and the keys and values come from the output of the perceptual module. This cross attention operation is followed by an FFN update to $\mathcal{I}$. Consider the perceptual module outputs for a chunk $l$ to be $\bar{X}_l = [\bar{x}_{l \times K}, \ldots, \bar{x}_{l \times K+K}]$. The update operation is implemented as follows:

$$\bar{\mathcal{I}} = \text{ATTENTION}(\text{LN}(\mathcal{I}_l), \text{LN}(\bar{X}_l), \text{LN}(\bar{X}_l)) + \mathcal{I}_l$$
$$\mathcal{I}_{l+1} = \text{FFN}(\text{LN}(\bar{\mathcal{I}})) + \bar{\mathcal{I}} \tag{4}$$

The temporal latent bottleneck module introduces the notion of recurrence in our model. We show the details of this module in Figure 1 (inside the circle).

**Perceptual Module + Temporal Latent Bottleneck Model.** We now present our complete architecture integrating both the perceptual module and the temporal latent bottleneck together. Given a sequence of tokens $X = [x_0, x_1, x_2, \ldots, x_t]$. We chunk this input into chunks of size $K$ resulting in $\lfloor T/K \rfloor$ chunks. The chunks are processed sequentially one after the other. For a chunk $k$, it is first processed using the perceptual module conditioned on information from the temporal latent bottleneck state. The outputs of the chunk are used to update the temporal latent bottleneck state $\mathcal{I}$. The resultant temporal latent bottleneck state is then used to process the next chunk. The full model is presented in Figure 1. We use a Transformer as the perceptual module in our experiments. Thus our main contribution is introducing a temporal latent bottleneck into Transformers and showing its advantages through a variety of experiments. We also present the detailed algorithm for the proposed approach in Algorithm 1.

The proposed model is similar to a parallel work called Block Recurrent Transformers [37]. There are few differences between our work and theirs. First, they use a sliding window attention, while

we divide the input into chunks. In their paper, they perform cross attention and self attention in parallel while we find that doing them sequentially and performing cross attention once per $R$ self attention steps yields better results. We defer the rest of the discussion on related works to Appendix Section 6

## 3  Experiments

Our goal is to show the wide applicability and benefits offered by the *temporal latent bottleneck*, which we refer to as TLB. We demonstrate that the proposed model outperforms competitive baselines across many domains including vision, reinforcement learning, and natural language. Our main goal is to show that the proposed approach has high expressive power like Transformers while also being sample efficient unlike Transformers. Thus our main baselines are based on the original Transformer architecture. For example, we compare against ViT [25] in image classification, Deci-

Table 1: **Image Classification**. Here we compare the performance of the proposed VIT + TLB model against VIT and SWINV2 on CIFAR10 and CIFAR100 datasets for $64 \times 64$ images and $128 \times 128$ images. Note that the model is trained only on the $64 \times 64$ sized images and then transferred to $128 \times 128$ sized images. Results averaged across 3 seeds.

| | CIFAR10 | | CIFAR100 | |
|---|---|---|---|---|
| MODEL | $64 \times 64$ | $128 \times 128$ | $64 \times 64$ | $128 \times 128$ |
| VIT | 93.75 | 73.18 | 69.53 | 47.4 |
| SWIN V2 | 97.66 | 84.9 | 79.95 | 58.59 |
| VIT + TLB | 94.79 | 84.38 | 79.17 | 59.19 |

sion Transformer [13] in Reinforcement Learning, and Vanilla Transformer in rest of the tasks. We also compare against some representative baseline that offer some of the key properties that our model offers. For example, we compare against state-of-the art Swin Transformer [50] which is a strong baseline for image classification and is also hierarchical similar to the proposed model. We also compare against Transformer LS [70] which also processes long-term and short-term information using different attention streams. Furthermore, we also compare against Feedback Transformer [26], which also introduces top-down communication into Transformers. Another key point of the proposed model is that any position cannot attend to any information from the future beyond its chunk since the temporal latent bottleneck only *summarizes the past, not the future*. Meanwhile, **all the baselines we consider have bidirectional context** i.e. they can attend to all of the past and the future. We observe that despite this limitation, the proposed model outperforms all the considered baselines.

### 3.1  Temporal Latent Bottleneck For Perception

**Image Classification.** Recently, Transformers have been widely applied for visual perception and have shown strong performance improvements over CNNs in tasks such as image classification, semantic segmentation, instance segmentation, etc. In this work we focus on image classification using Transformers. For a model to do well on image classification, it should learn to only focus on the relevant information and ignore other details (eg. background information). Self attention does not inherently have this inductive bias of ignoring irrelevant information since it models all pairwise interactions between the inputs. We posit that adding a limited bandwidth temporal latent bottleneck into

Table 2: Here we show the performance of the proposed ViT + TLB model against two baselines - One with no access to the past and One with no top-down information (i.e. high level to low level communication). We can see that the model suffers a drop in performance for both the baseline thus showing the importance of past information and top-down communication. Results averaged across 3 seeds.

| | | | CIFAR10 | |
|---|---|---|---|---|
| MODEL | PAST INFO | TOP DOWN | $64 \times 64$ | $128 \times 128$ |
| VIT + TLB | ✓ | ✓ | 94.79 | 84.38 |
| NO PAST INFO. | × | × | 91.30 | 72.92 |
| NO TOP-DOWN CONDN | ✓ | × | 93.75 | 83.59 |

the Transformer will allow the model to focus only on the most important information in the image which should enable the model to perform well.

**Results**. We test our hypothesis on the CIFAR10 and CIFAR100 [47] image classification datasets. We also test the generalization abilities of the models by comparing their performance on images of higher resolution ($128 \times 128$) than seen during training ($64 \times 64$). We use ViT [25] and Swin Transformer V2 (denoted as Swin V2) [50] as our baselines. Swin Transformer V2 has a key strength of generalizing to higher resolution images than those seen during training, making it a strong baseline. The input image is split into patches of size $4 \times 4$ and fed in rastor order to all the models. For the proposed model we use ViT as the perceptual module and add a temporal latent bottleneck module

to it. We call this model VIT + TLB. To predict the classification scores, we take the mean across the final temporal latent bottleneck state vectors and pass the resulting representation through an MLP. We present the results for this experiment in table 1. *We can see that* VIT + TLB *outperforms* VIT *for all cases and performs competitively to Swin Transformer V2.* For further hyperparameter details, we refer the reader to Appendix section 7.1.

**Quantitative Analysis**. One essential component of our model is top-down conditioning. Top down information helps in integrating information from the past as well as high-level information into the perceptual module. We hypothesize that both these kinds of information are important for the model to perform well. To test this, we design two baselines - (1) VIT + TLB (NO PAST INFO): In this baseline, we do not allow the TLB to communicate to the perceptual module, therefore the perceptual module has no information about the past; (2) VIT + TLB (NO TOP-DOWN CONDN): In this baseline, we have a separate temporal latent bottleneck module at every layer, therefore the perceptual module has access to past information but does not have access to any high-level information through top-down feedback. We show the results for this ablation in Table 2. We can see that the performance of both the baselines is worse than the proposed VIT + TLB model. This shows that both high-level information through top-down feedback and information from the past is important for the model to perform well.

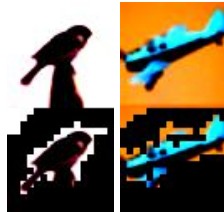

Figure 2: We want to analyze the information which is being used to influence the processing of Temporal Latent Bottleneck. To calculate this, we calculate the attention scores for different patches and we mask out all the patches that are not in the top 30% of the attention scores. We can see that for both the images it recovers the foreground almost perfectly which shows it learns to focus on the most important information required to solve the downstream task.

**Qualitative Analysis**. To get a better understanding of what the temporal latent bottleneck is doing, we visualize the parts of the image where the temporal latent bottleneck pays most attention while it is being updated by the perceptual module. We present this visualization in Figure 2. We can see that the temporal latent bottleneck learns to pay the most attention to the foreground in each case. This further confirms our hypothesis that the limited capacity bottleneck focuses on the most important information required to solve the downstream task.

**Self Supervised Learning.** Many recent works have used Vision Transformers for self-supervised learning [8, 2, 32, 12, 49, 48]. Here we show a proof-of-concept that introducing a temporal latent bottleneck in Vision Transformers results in better self-supervised representations. We consider the SiT model from [2] for this experiment. They use 3 objectives to pretrain their model - (1) The Reconstruction Objective - Reconstructs the input image, (2) The Rotation Prediction Objective - Predicts the rotation angle from $[0°, 90°, 180°, 270°]$, and (3) The Constrastive Objective (similar to SimCLR [14]). For the proposed approach, we introduce a temporal latent bottleneck into SiT resulting in the SiT + TLB model. SiT also uses additional trainable contrastive and rotation tokens as input for calculating the contrastive and rotation objectives respectively. For SiT + TLB, we

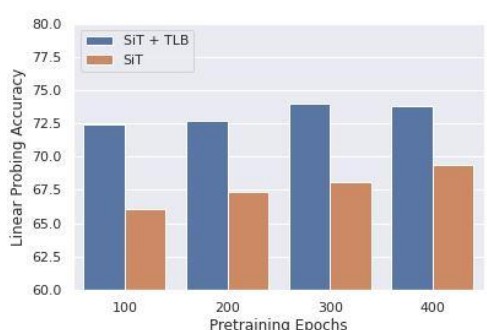

Figure 3: **Self Supervised Learning** Results of linear probing on the CIFAR 10 dataset for models pretrained on the STL 10 dataset. We can see that the proposed SiT + TLB approach outperforms SiT.

take the mean across the temporal latent bottleneck state vectors and use the resulting representation for computing the rotation and contrastive objectives. We use a chunk length of 20 for the SiT + TLB model. We pretrain the model for 400 epochs and evaluate the pretrained model at different epochs using linear probing.

**Results**. To evaluate the model, we pretrain the model on the STL10 dataset [22] and evaluate the learned representation using linear probing on the CIFAR10 dataset [47]. We present the re-

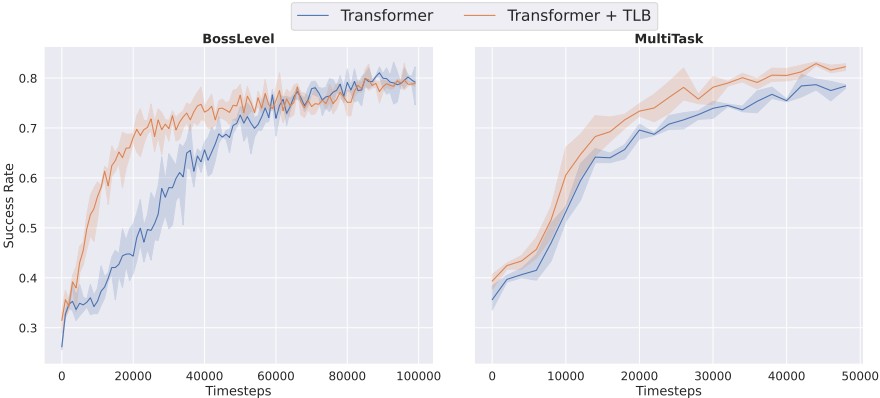

Figure 4: **Single Task BabyAI**. (Left) Here we compare the performance of Transformer and Transformer + TLB on the BossLevel task from BabyAI. We can see that while both the models converge to a similar success rate, Transformer + TLB converges faster than Transformer. **Multi Task BabyAI**. (Right) Here we compare the performance of Transformer and Transformer + TLB on 8 tasks from the BabyAI suite of environments. A single model is trained for all the 8 tasks. We can see that Transformer + TLB converges faster and achieves a better performance than Transformer.

sults for this experiment in Figure 3. We can see that the proposed approach outperforms SiT thus showing the effectiveness of the proposed architecture for self-supervised learning. For additional experimental results and details, we refer the reader to Appendix section 7.2.

## 3.2 Temporal Latent Bottleneck for Sequential Decision Making

Transformers have recently been used for sequential decision making in reinforcement learning tasks such as Atari and BabyAI [13, 38]. These works deploy Transformers in the offline RL setting where a large number of trajectories are available either through another trained agent or an expert agent. The Transformer is trained as an autoregressive generative model that predicts actions conditioned on the past context. We incorporate the temporal latent bottleneck module into the Transformer and explore its benefits in the RL setting. We test the proposed model in the BabyAI [15] and Atari [9] benchmarks. We describe our setups in detail below.

**Instruction Based Decision Making: BabyAI.** BabyAI [15] provides a suite of environments where the agent has to carry out a given instruction in a partially-observable maze. These instructions include competencies such as going to an object in the maze, placing an object beside another object in the maze, opening a door with a key, etc. Some environments in the benchmark contain instructions that combine multiple competencies sequentially. For example, *pick up a red ball and open the door in front of you after you pick up the grey ball on your left and pick up a red box*. Each environment in Baby AI benchmark has a different type of instruction that tests a different competency. The *BossLevel* is the most complicated environment that contains instructions from all competencies. For more details regarding the various environments from the BabyAI benchmark, we refer the reader to Appendix section 7.4.

We train our models with behavior cloning using expert trajectories from an oracle. For evaluation, we test the model by directly deploying it in the environment. We report the *success rate* which measures whether the agent successfully carried out the given instruction or not. We use a Transformer [61] as the baseline in these experiments. For the proposed model, we introduce a temporal latent bottleneck into the Transformer-based perceptual module. For the baseline Transformer model, we append the language instruction to the sequence of states allowing the model to attend to the language instruction at each layer. For the proposed model, the language instruction is appended to each chunk, allowing each chunk to attend to it.

**Results**. We consider two settings - **Single task** and **Multi task**. In the single task setting, we evaluate the proposed approach on individual environments from the BabyAI benchmark while in the multi-task setting we train a single model on 8 different environments.

**Single Task.** We present the results for BossLevel in Figure 4 (left) and present the results for the other tasks in Appendix Figure 9. *We can see that while Transformer and Transformer + TLB achieve almost similar performance at convergence. However, Transformer + TLB is much more sample efficient, converging much faster*. We posit that the temporal latent bottleneck module prohibits the model from paying attention to unnecessary information which allows it to converge faster.

**Multi Task.** We present the results for the multi task setting in Figure 4 (right). We train the model on 8 environments - PutNext, Unlock, Synth, GoToSeq, SynthLoc, GoToImpUnlock, BossLevel. We evaluate the model on the same 8 environments. We report the average success rate across 8 games. *We can see that the Transformer + TLB model converges faster and also outperforms the Transformer*. We refer the reader to the appendix for more details regarding the model and training.

**Atari.** [13] recently introduced the Decision Transformer (DT) which learns to play various games in the Atari benchmark from suboptimal trajectories of a learned agent. Decision Transformer models the offline RL problem as a conditional sequence modelling task. The model uses a causal mask and supervised training to match the actions in the offline dataset conditioned on the future expected returns and the past history. This is done by feeding into the model the states, actions, and the return-to-go

Table 3: **Atari**. Here we show that adding a temporal latent bottleneck into decision Transformer improves performance across various atari games. Results are averaged across 10 seeds.

| GAME | DT | DT + TLB |
|---|---|---|
| BREAKOUT | $71.51_{\pm 20.58}$ | $87.63_{\pm 16.24}$ |
| PONG | $13.68_{\pm 2.00}$ | $14.71_{\pm 1.78}$ |
| QBERT | $3268_{\pm 1773.07}$ | $5019.75_{\pm 1647.13}$ |
| SEAQUEST | $1039.11_{\pm 122.90}$ | $1248.22_{\pm 86.62}$ |

$\hat{R}_c = \sum_{c'=c}^{C} r_c$, where $c$ denotes the timesteps. This results in the following trajectory representation: $\tau = \left( \hat{R}_1, s_1, a_1, \hat{R}_2, s_2, a_2, \hat{R}_3, s_3, a_3, \dots \right)$, where $a_c$ denotes the actions and $s_c$ denotes the states. At test time, the start state $s_1$ and desired return $\hat{R}_1$ is fed into the model and it autoregressively generates the rest of the trajectory. Experimental results show that DT can leverage the strong generalization capabilities of Transformers and achieve the desired returns in a wide variety of tasks in Atari and OpenAI Gym, outperforming previous approaches in offline RL.

We use the same setup as used in [13] for our experiments. We set the context length to a fixed number $C$. During training, $C$ timesteps from an episode are sampled and fed into the model resulting in a trajectory of length $3C$ (considering 3 modalities - returns-to-go, states, and actions). Each modality is processed into an embedding of size $d$. The state is processed using a convolutional encoder into an embedding of size $d$. The resulting trajectory is fed into the decision Transformer. The outputs corresponding to the states $s_c$ are fed into a linear layer to predict the action $a_c$ to be taken at timestep $c$. For the proposed model, we incorporate a temporal latent bottleneck module into the Decision Transformer.

Table 4: **Long Range Dependencies**. Here we compare the performance of the proposed model against the recently proposed long-short Transformer model [70] and the vanilla Transformer model [61]. We can see that the proposed model outperforms both the baselines thus showing the superiority of the proposed model in modelling long-range and hierarchical dependencies. Results averaged across 5 seeds.

| MODEL | LISTOPS | TEXT CLASSIFICATION |
|---|---|---|
| TRANSFORMER | $37.64_{\pm 0.0001}$ | $64.0_{\pm 0.0001}$ |
| TRANSFORMER LS | $37.5_{\pm 0.0002}$ | $65.5_{\pm 0.0003}$ |
| TRANSFORMER + TLB | $38.2_{\pm 0.0001}$ | $82.08_{\pm 0.44}$ |

**Results**. We present our results in Table 3. The model is trained on 1% of the Atari DQN-replay dataset [1] (500K transitions for each game). We use the same 4 games used in [13]: Pong, Seaquest, Qbert, and Breakout. *We can see that the proposed model outperforms Decision Transformer in all the considered games thus showing the effectiveness of the proposed model*. More details regarding the model and training can be found in the appendix section 7.5.

### 3.3 Temporal Latent Bottleneck for Long Range Dependencies

Here, we test the effectiveness of the proposed model in modelling long range dependencies. We apply the proposed model on the ListOps and text classification tasks from the Long Range Arena (LRA) benchmark [60]. Both these tasks have very long sequences ranging from 1K to 4K tokens. Thus, for a model to do well, it has to learn to capture dependencies across very long time scales. Additionally, all these tasks have an inherent hierarchical structure. For example, Listops consists of nested list operations which makes it hierarchical. For text classification, the inputs consist of text in the form of bytes. Therefore, the model has to learn to compose bytes into characters and characters into words. We hypothesize that the multi-scale hierarchical nature of the proposed model will be extremely useful in modelling such hierarchical information.

**Results**. For this experiment, we use the same setup as[70]. For the proposed model, we use a Transformer as the perceptual model and implement the temporal latent bottleneck as described in Section 2.2. We take the mean across the temporal latent bottleneck state vectors and use the resulting representation for classification. We compare the model against the long-short Transformer (LS) model [70], which is a recently proposed model for the long range arena benchmark, and the vanilla Transformer model [61]. We present the results in Table 4. *We can see that the proposed model outperforms both the baselines in both the tasks thus showing its usefulness in modeling long range dependencies*. For further details, we refer the reader to Appendix section 7.3.

In Fig. 5, we plot the convergence curves for ListOps and Text Classification. For ListOps (Figure 5(a)), we plot the convergence curves against the number of samples i.e. we do only one pass over the dataset hence the model does not see any example more than once. We can see that the proposed Transformer + TLB model is much more sample efficient than the baseline Transformer LS model. For Text Classification (Figure 5(b)), we plot the convergence curves against the number of training steps. We find that doing only one pass over the dataset does not work well for both the baseline and the proposed model hence we use number of training steps on the x-axis. We can see that while initially both models converge at a similar pace, the proposed model achieves a much higher performance.

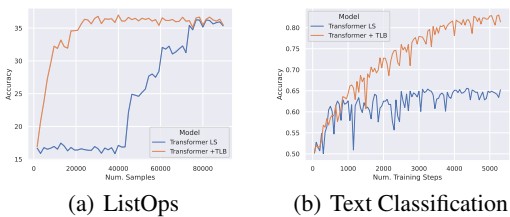

(a) ListOps      (b) Text Classification

Figure 5: **(a)** Here we show the performance on ListOps as a function of the number of samples in the dataset. We do only one pass over the entire data and find that Transformer +TLB takes much fewer samples to converge as compared to the baseline Transformer LS. **(b)** Here we show the convergence curves of both the Transformer + TLB model and the Transformer LS model on the text classification task. In this case, we do not perform only one pass over the dataset since we observe that both models do not reach convergence in a single pass. Therefore, we report the number of training steps on the x-axis. We can see that the proposed model achieves much higher score than the baseline.

We measure the wall-clock time and memory required for text classification task as we vary the chunk size in Table 5. All TLB models have an increased memory efficiency and supports faster inference speeds with respect to the baseline transformer model. The training speeds also get better with increased chunking. The only exception is very small chunk sizes, where the training is slower than the baseline because of increased temporal unrolling. However, as shown in Figure 5, such models are very sample efficient resulting in lesser training steps overall.

Table 5: **Text Classification - Performance Ablation** Here, we compare the wall-clock time and memory during the training and inference phase of the text classification task w.r.t baseline transformer model.

| CHUNK SIZE | 1000 | 100 | 40 | 20 | 10 |
|---|---|---|---|---|---|
| INFERENCE SPEED | 3.5x | 3.6x | 3.3x | 2.2x | 1.2x |
| INFERENCE MEMORY | 0.09x | 0.08x | 0.12x | 0.08x | 0.1x |
| TRAINING SPEED | 4.4x | 4.4x | 2.2x | 1.4x | 0.7x |
| TRAINING MEMORY | 0.14x | 0.08x | 0.49x | 0.40x | 0.42x |

**Analysis**. Here we perform an ablation to show that the Temporal Latent Bottleneck does not only contain short-term information but also summarizes information from long term past. To test this hypothesis we design a baseline in which the current chunk attends to the previous few chunks instead of attending to the temporal latent bottleneck. We find that this baseline achieves a performance of $32.10_{\pm 0.019}$% compared to the proposed models $38.2_{\pm 0.0001}$% on the ListOps task. This shows that the Temporal Latent Bottleneck contains information about the long-term past. Additionally, here also we perform an experiment to probe the importance of top-down communication (i.e. high level to low level feedback). To do this we use the same Transformer + TLB (No Top-Down Condn) baseline used in Table 2. We find that this baseline achieves a performance of $37.57_{\pm 0.003}$% which is lower than the performance of the proposed Transformer + TLB model which

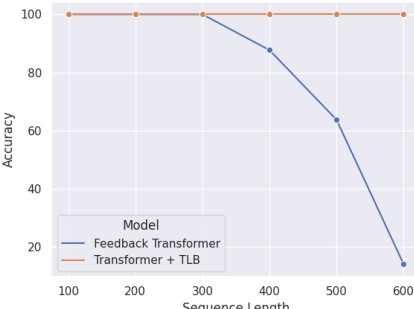

Figure 6: **Copying Task**. Here we compare the performance of the proposed Transformer + TLB model to the Feedback Transformer model on the copying task. We can see that the Transformer + TLB achieves perfect accuracy for all the studied sequence lengths while the the performance of Feedback Transformer starts dropping after sequence length 400.

| Sequence Length | Feedback Transformer | Transformer + TLB |
|---|---|---|
| 100 | 11800 | 6200 |
| 200 | 16600 | 9100 |
| 300 | 35100 | 12700 |
| 400 | NA | 14600 |
| 500 | NA | 13600 |
| 600 | NA | 19300 |

Table 6: **Copying Sample Efficiency Ablation**. Here we present the number of unique samples required for the models to reach to perfect accuracy on the copying task. NA indicates that the model does not reach perfect accuracy. We can see that in all cases the Transformer + TLB model is more sample efficient than the Feedback Transformer model.

achieves $\mathbf{38.2}_{\pm 0.0001}\%$ which further shows that top-down information from high-level to low-level is important for the model to perform well.

We perform additional experiments to give us more insight into the behavior of the proposed model. We present these experiments in Appendix Section 7.3. We also compare the model to additional efficient transformer baselines for all LRA tasks in Appendix Table 9.

**Temporal Latent Bottleneck for Copying Task.** Here, we study the copying task used in [36]. In the copying task, the model receives a sequence of 10 digits followed by blank inputs for a large number of steps, and then the model is asked to output the sequence of digits it received initially. Therefore, the model has to remember the original sequence of digits across long time scales. We can control the sequence length of this task by controlling the length of the blank input.

The main motive behind studying this task is comparing the model to the Feedback Transformer model introduced in [26] which also has top-down attention similar to the proposed model but does not represent information at multiple scales. We compare both the models on the copying task for sequence lengths 100, 200, 300, 400, 500, and 600. We present the results for this task in Figure 6. We can see that while both Transformer + TLB and Feedback Transform perform well for low sequence lengths, the performance of Feedback Transformer drops for longer sequence lengths above 400 while the proposed Transformer + TLB model still achieves perfect accuracy at long sequence lengths. We also compare the sample efficiencies to achieve perfect accuracy for both the models. We present this result in Table 6. We can see that the proposed Transformer + TLB is more sample effecient than the baseline Feedback Transformer achieving perfect accuracy in much lesser number of samples in each case. For further details we refer the reader to Appendix Section 7.6.

## 4    Conclusion

We have developed an approach aimed at introducing selectivity in the interactions across time-steps in a transformer by splitting processing into two streams: (a) a slow stream that is updated in a recurrent manner and (b) a fast stream that processes the visual input. The two streams are parameterized independently and interact with each other via attentional bottleneck. The information processed by the fast stream is used to change the state of the slow stream, and the information in the slow stream is used by the fast stream as contextual information to process the input. Through our experiments we show that the proposed approach works well across wide range of domains and problems. One limitation of the proposed model is that the chunk size is fixed and treated as a hyperparameter which requires some domain knowledge. Future work should explore methods for dynamic chunking.

## 5    Acknowledgement

The authors would like to thank Compute Canada for providing the computational resources used in this project. The authors also gratefully acknowledge the funding from Samsung, IBM and CIFAR.

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
