# OpenReview forum: "Temporal Latent Bottleneck: Synthesis of Fast and Slow Processing Mechanisms in Sequence Learning"
_NeurIPS.cc/2022/Conference — NeurIPS 2022 Accept_

### Official Review · Reviewer_CN5N · 2022-07-08

**Rating:** 5
**Confidence:** 4
**Soundness:** 2 fair
**Presentation:** 3 good
**Contribution:** 1 poor

**Summary:**

The authors introduce a transformer-based architecture that processes sequences in sequential chunks or blocks. Each chunk includes K elements from the sequence and is processed by a stack of transformer layers that include self-attention and cross-attention heads. Then the output of each chunk is processed sequentially using another cross-attention-based transformer. The hidden or latent state generated at output of this layer, is included as input to the cross-attention layers in the subsequent chunk. The method is then compared against transformers in image classification, self-supervised learning, reinforcement learning, and language tasks. The paper exhibits improved results across all tasks.



**Questions:**

Questions to the authors:
* How is the network trained given the recurrent connections?
* How is the network initialized? In particular, how is the initial $I$ selected?
* What ways the input can be partitioned for chunking? Why the one you choose is the best? How does the chunking influence the quality of the network? Can it always be applied?
* What are the model sizes? Is there any difference when using the recurrent model presented here versus the other transformer models?
* How does your method compare to [22], [53] (and other similar papers) in the experiments you presented?
* Do you use "bidirectional" (or non-causal mask) for language modeling within each chunk?

**Limitations:**

Societal impacts are not described in the paper.
Limitations: What would happen if the critical input data is "chunked" between two related pieces of information? Would the model be able to solve it at the same quality of a fully parallel transformer?
What are the pros and cons in the runtime to this method compared to the vanilla transformer?


**Strengths And Weaknesses:**

The idea introduced in the paper of mixing recurrent networks using attention and transformers is not novel. It has been presented in several works in different flavors. Referenced papers [22] and [53] are two examples. Authors would like to check [a,b,c,d] below and explain how novel their solution is compared to these. The novelty of the proposed method seems limited.

The paper is of empirical nature. The experiments compare results of the proposed architecture with other transformers that are not necessarily the state of the art. It is unclear whether there is a benefit to this method when comparing to more advanced transformers that aim to resolve similar problems. The generalization results from resolution of 64 to 128 pixels is nice but not sufficient (e.g., the results still drop by 20% in accuracy for CIFAR100). Some works show other possibilities for generalization. Other analysis beyond the ablation of the "top-down" connection would have been relevant to show the value of this method.

The notation in the paper is confusing for moments, it could be more consistent. The experiments section is missing many details in the main text.

The work is relevant to this community. However, given the limited novelty, its significance is low.

[a] Recurrent Independent Mechanisms, Goyal et al., ICLR 2021
[b] Block-Recurrent Transformers, Hutchins et al., 2022
[c] MART: Memory-Augmented Recurrent Transformer for Coherent Video Paragraph Captioning, Lei et al., ACL 2020
[d] Transformers are RNNs: Fast Autoregressive Transformers with Linear Attention, Katharopoulos et al., ICML 2020


*************** UPDATE *******************
I updated the score to reflect the additional results presented by the authors during the rebuttal.

---

> ### Author Response · Authors · 2022-08-02
> **Regarding Novelty**
>
> We thank the reviewer for the detailed review and the constructive feedback.
>
> We would first like to give a general update that we were able to improve the performance of the proposed model on the text classification task from the long range arena benchmark. We now achieve a performance of 82.08% as compared to 67.8% that we reported in the submission. This beats the highly competitive S4 model [1] which achieves 76.02%. The main improvement comes from reducing chunk size to 10 (+6%) and adding local positional embedding within each chunk (+8%).
>
> **Regarding Novelty**
>
> Here, we explain the novelty of the paper w.r.t the 4 paper pointed by the reviewer.
>
>  - RIMs [2] RIMs introduce the idea of having a set of slots, where each slot attends to different parts of the region using "cross-attention". In RIMs, there's no inductive bias towards  having multiple-streams of computation operating at different time-scales, and having information asymettry between the two. In our work, fast module operates on fast time-scale and the TLB module operates on slower time-scale.
>
>
>
>  - Transformers are RNNs: Fast Autoregressive Transformers with Linear Attention, [5] This paper express the self-attention as a linear dot-product of kernel feature maps and make use of the associativity property of matrix products to reduce the complexity of attention from quadratic to linear in th number of positions. We have added the rsults for LRA showing the proposed work outperforms linear attention. There are two important differences (a) absence of multiple streams of computation operating at different time-scales; (b) informtion asymmetry leading to effective compression of the learned representations. We also compare the performance of the proposed model to this model on the long-range arena benchmark. We obtain the following results:
>
>      | Model | ListOps | Text Classification |
>      | ----- | ------- | ------------------- |
>      | Linear Attention [5] | 16.13 | 65.90 |
>      |Transformer + TLB | 38.2 | 82.08 |
>
>  - Block Recurrent Transformers [3]: As mentioned in section 2 of the paper, Block Recurrent Transformer is work done in parallel to our work. While there are many similarities between our work and theirs, there are also some differences.  First, they use a sliding window attention, while we divide the input into chunks. In their paper, they perform cross attention and self attention in parallel while we find that doing them sequentially yields better results. We compare the parallel and sequential paradigms for doing cross attention and self attention on the CIFAR 10 dataset. We get the following results:
>
>     | Model | 64 x 64 | 128 x 128 |
>     | ----- | ------- | --------- |
>     | Sequential (Ours) | 94.79 | 84.38 |
>     | Parallel (Block Recurrent Transformer) | 85.42 | 66.93 |
>
>     Thus, we can see that our way of doing sequential cross and self attention yields better results than the parallel way proposed in Block Recurrent Transformers. Another difference, between our work and theirs is that in section 3.5 of the Block Recurrent Transformer paper [https://arxiv.org/abs/2203.07852], the authors say that "the training stability is quite sensitive to how the gates are initialized". To mitigate this issue they initialize the bias to small non-zero values. Also, they "add a constant of -1 and +1 to the input and forget gates (see Eq. 4) to initially bias the gate to “remember”". These are the special tricks that we refer to. The proposed model does not need any such tricks for performing well and we do not observe any training stability issues (we will open source our code). Also, while their focus in mainly on natural language processing tasks, we focus on a much broader variety of tasks covering many different domains.
>
>
> [1] Efficiently Modeling Long Sequences with Structured State Spaces Gu et al 2021
>
> [2] Recurrent Independent Mechanisms, Goyal et al., ICLR 2021
>
> [3] Block-Recurrent Transformers, Hutchins et al., 2022
>
> [5] Transformers are RNNs: Fast Autoregressive Transformers with Linear Attention, Katharopoulos et al., ICML 2020

---

> > ### Author Response · Authors · 2022-08-02
> > **Regarding Novelty Contd**
> >
> > - MART: Memory-Augmented Recurrent Transformer for Coherent Video Paragraph Captioning [4]: This is relevant work. We have cited this work in the main paper. Two big differences: (a) [4] does attention among tokens, and cross-attention to the memory M in parallel similar to Block Recurrent Transformer, where-as we found that doing them sequentially  improves the results (as shown in the above table), (b) [4] has separate memory tokens for each layer. Each layer attends and writes to its own memory tokens, and at each time-step. On the contrary, we just have one set of memory tokens (which we call TLB) and we write to the TLB only ones, and each layer reads from the same TLB. We did more experiments comparing the proposed model to a baseline in which we have a separate TLB (or memory tokens) at each layer. We evaluated this baseline on the byte-level text classification task, the ListOps task, and CIFAR10 Image Classification. We find that in all cases this baseline performs worse than the proposed model:
> >
> >     - CIFAR10 Image Classification - ViT + TLB: 94.79%, MART-Style Baseline : 93.75%
> >     - ListOps: Transformer + TLB: 38.2%; MART-Style Baseline: 37.57%
> >     - IMDB Text Classification - Transformer + TLB: 82.08%; MART-Style Baseline: 80.68%
> >
> >      We also show the utility of the proposed architecture in various different problems ranging from language modelling, self-supervised learning, imitation learning, perceptual vision problems as compared to just video captioning in [4].
> >
> > [4]  MART: Memory-Augmented Recurrent Transformer for Coherent Video Paragraph Captioning, Lei et al., ACL 2020

---

> > > ### Author Response · Authors · 2022-08-02
> > > **Regarding Experimental Results**
> > >
> > > **Regarding Experimental Results**
> > >
> > > The reviewer suggested that the proposed method has not been compared with the state of the art transformer architectures. We thank the reviewer for bringing this up. However, to the best of our knowledge, we compare the proposed model the best performing models in each task as follows:
> > >
> > > 1) Image Classification: To the best of our knowledge Swin transformer [6] is one of most competive transformer-based models for image classification. Swin transformer v2 [7] is an upgrade on the original Swin transformer and was specifically designed to have strong resolution extrapolation capabilities. To the best of our knowledge Swin Transformer v2 is the state-of-the-art for generalization to larger resolution images.
> > > 2) The long-range arena benchmark: To the best of our knowledge, long-short transformer [8] is one of the best performing transformer-based baselines. In the paper it is shown to outperform vanilla transformer [9], Nystromformer [10], Performer [11], Linformer [12], and Reformer [13]. Additionally, the proposed Transformer + TLB model beats almost all transformer-based baselines for the listops and text classification tasks mentioned here https://paperswithcode.com/sota/long-range-modeling-on-lra except PSF [14] and Paramixer [15] to which we achieve competitive performance.
> > >
> > >     | Model | ListOps | Text Classification |
> > >     | ----- | ------- | ------------------- |
> > >     | PSF   | 38.85%|77.32% |
> > >     |Paramixer | 39.57% | 83.32% |
> > >     | Transformer + TLB | 38.20 % | 82.08 % |
> > >
> > > 3) Sequential decision making in an offline setting: Most recent works such as Decision Transformer [16], Trajectory Transformer [17], GATO [18], and many other works mainly contribute different formulations to frame the reinforcement learning problem as a sequnce modeling problem but the core architecture that they use is still a vanilla transformer. Therefore, we believe vanilla transformer (i.e., Decision Transformer) is a competitive baseline for offline sequential decision making problems.
> > >
> > > However, there is a possibility we may have missed works that outperform the above baselines in each of the above settings. If the reviewer has any such suggestions that they could point us to, we would be happy to perform experiments against that model.
> > >
> > > [6] Swin Transformer: Hierarchical Vision Transformer using Shifted Windows liu et al 2021
> > >
> > > [7] Swin Transformer V2: Scaling Up Capacity and Resolution liu et al 2021
> > >
> > > [8] Long-Short Transformer: Efficient Transformers for Language and Vision
> > >  zhu et al 2021
> > >
> > > [9] Attention is all you need vaswani et al 2017
> > >
> > > [10] Nyströmformer: A Nyström-Based Algorithm for Approximating Self-Attention Xiong et al 2021
> > >
> > > [11] Rethinking Attention with Performers Choromanski et al 2020
> > >
> > > [12] Linformer: Self-Attention with Linear Complexity wang et al 2020
> > >
> > > [13] Reformer: The Efficient Transformer Kitaev et al 2020.
> > >
> > > [14] Sparse Factorization of Large Square Matrices Khalitov et al 2021
> > >
> > > [15] Paramixer: Parameterizing Mixing Links in Sparse Factors Works Better than Dot-Product Self-Attention yu et al 2022
> > >
> > > [16] Decision Transformer: Reinforcement Learning via Sequence Modeling Chen et al 2021
> > >
> > > [17] Reinforcement Learning as One Big Sequence Modeling Problem Janner et al 2021
> > >
> > > [18] A Generalist Agent Reed et al 2022

---

> > > > ### Author Response · Authors · 2022-08-02
> > > > **Comparison to Feedback Transformer and More Analysis**
> > > >
> > > > As asked by the reviewer, we also compare the proposed model against Feedback Transformer [19]. We use the copying task for this comparison. In this task the model is first given a sequence of 10 digits as input and it is supposed to output the same sequence after an interval of blank inputs. We describe the task in detail in Section 3.4 of the paper. The length of the blank inputs in the middle can be as high as needed resulting in very long sequence lengths. For a model to do well, it needs to remember the original sequence across long time-scales. We compare the proposed Transformer + TLB model to the Feedback Transformer [19] which has been shown to work well on this task. We compare both the models for varying sequence lengths from 100 to 600 as shown in Figure 5 in the main paper. We see that for higher sequence lengths the performance of the Feedback Transformer starts dropping while the proposed model still achieves perfect accuracy. This shows that the TLB retains the original sequence across long time scales without overwriting any of its information in the presence of blank inputs. The Feedback Transformer is unable to do so.
> > > >
> > > > We apologise to the reviewer for not being able to show the results against BRIMs [20]. We tried the BRIMs model on our CIFAR10 classification setup but were not able to get it to converge.
> > > >
> > > > [19] Adressing some limitations of Transformer with Feedback Memory Fan et al 2021
> > > >
> > > > [20] Learning to combine top-down and bottom-up signals in recurrent neural networks with attention over modules Mittal et al 2020
> > > >
> > > > **On Requiring more Analysis**
> > > >
> > > > We have conducted more experiments analysing the proposed model and its various components. We have updated the main paper with these experimental results. Here, we provide a brief overview of our analysis experiments -
> > > >
> > > > *Impact of Top-Down Information*
> > > >
> > > > The TLB is not only important because it provides information from the past but also because it provides high-level information through top-down communication. We claim this top-down communication is important for the model to perform well. To confirm our claim, we design a baseline which is similar to our proposed Transformer + TLB model but it has a different TLB module after each layer similar to MART [4] as opposed to having only one TLB module after the last layer. Thus, each layer L of the perceptual module reads and writes information from the TLB module specific to layer L. Therefore, in this case the TLB module only provides information about the past, it does not contain any top-down information since the TLB module at layer L does not communicate with the lower layers of the perceptual module. We evaluate this baseline on CIFAR10 image classification, IMDB text classification, and ListOps. The results are as follows:
> > > >
> > > > - CIFAR10 Image Classification - ViT + TLB: 94.79%, ViT + TLB (No Top-Down): 93.75%
> > > > - ListOps: Transformer + TLB: 38.2%; Transformer + TLB (No Top-Down): 37.57%
> > > > - IMDB Text Classification - Transformer + TLB: 82.08%; Transformer + TLB (No Top-Down): 80.68%
> > > >
> > > > This shows that the TLBs ability to provide top-down information along with information from the past is important for the model to perform well. Only providing summarized information from the past is not enough.
> > > >
> > > > *TLB Captures the most important information required to solve the downstream task*
> > > >
> > > > Our hypothesis here is that the limited capacity of the TLB allows it to focus on the most important information in input required to solve the downstream task ignoring everything else that is not relevant. To confirm our hypothesis we perform two experiments:
> > > >
> > > > - First, for the image classification experiment, we visualize the patches that the TLB pays most attention to. To do this we first store the attention scores that the TLB pays to each patch in the image when the perceptual module is writing information to it (i.e. attention scores computed in the last step of Algorithm 1 in the Appendix). We only keep those patches in the image that fall in the top 30% of the attention scores masking everything else. We present the result of this visualization experiment in Figure 2 in the main paper. We can see that the TLB pays most attention to foreground of the image which shows that the TLB focuses on the most important information in the image learning to ignore irrelevant information.
> > > > - Second, our experiments on the copying task show that TLB retains the original sequence across long time scales without overwriting any of its information in the presence of blank inputs which shows that it captures and retains the most important information for solving the task.
> > > >
> > > > These two experiments confirm that the TLB learns to capture and retain important information without being affected by noisy or uninformative inputs.

---

> > > > > ### Author Response · Authors · 2022-08-02
> > > > > **More Analysis (Contd) and Answering Reviewer Questions**
> > > > >
> > > > >
> > > > > *TLB retains information across long time-scales*
> > > > >
> > > > > One of our main hypothesis is that TLB acts a long-term memory that stores information across long time scales. We already confirm this hypothesis by showing strong performance gains on task that require long-term contexts such as IMDB byte-level text classification (sequence lengths of ~4k), ListOps (sequence lengths of ~1k), and copying (sequence lengths of upto 600). To further give more concrete proof of this hypothese we design a baseline in which each chunk attends to only past few chunks instead of attending to the TLB. If this baseline achieves almost similar performance to TLB, this would show that the TLB only contains information from the recent few chunks. We evaluate this baseline on the ListOps task and find that it achieves an accuracy of 32.10% which is much lower than that of the proposed model Tranformer + TLB model which achieves 38.2%. This shows that the TLB caputures longer term information than the past few chunks.
> > > > >
> > > > >
> > > > > *Effect of the chunk size and number of TLB state vectors*
> > > > >
> > > > > The chunk size and number of TLB state vectors are hyperparameters. We vary the value these hyperparameters and study their effect on performance of the model.
> > > > >
> > > > > - **Chunk Size**: We try the following values for chunk size on the ListOps task: 2, 20, 100, 500, 1000. We present the results for this analysis in Appendix Table 6. We find that the model achieves best performance at chunk size 100, above or below which the performance drops.  Lower chunk sizes can potentially lead to a lot of information to integrate across chunks which might make the TLB forget important information more quickly and higher chunk sizes can lead to too much information to write in one chunk which can also lead to unwanted forgetting. We also evaluate various chunk sizes for the multi-task BabyAI experiment (Appendix Figure 9) and reach a similar conclusion that there is an optimal chunk size above or below which the performance drops.
> > > > > - **Number of State Vectors**: We try the following values for the number of state vectors on the ListOps task: 1, 10, 20, 200. Here also we find that there is an optimal number of state vectors above or below which the performance drops. Less number of state vectors can lead to very low capacity in the Temporal Latent Bottleneck leading to loss of important information. Similarly, a high number of state vectors can lead to a high capacity in the Temporal Latent Bottleneck leading to it capturing a lot of unnecessary of noisy information.
> > > > >
> > > > >
> > > > > **Questions raised by the reviewer**
> > > > >
> > > > > - How is the network trained given the recurrent connections?
> > > > >
> > > > >   The network is trained like a normal Transformer or RNN using backpropagation through time (BPTT)
> > > > > - How is the network initialized? How is the initial I selected?
> > > > >
> > > > >   We follow the default pytorch initialization for initializing the network i.e. its parameters are sampled from normal distributions. We initialize I by sampling its values from a normal distribution => torch.randn(batch_size, num_tlb_state_vectors, tlb_dimension)
> > > > > - What ways the input can be partitioned for chunking? Why the one you choose is the best? How does the chunking influence the quality of the network? Can it always be applied?
> > > > >
> > > > >   Currently, we only explore static chunking i.e. we have a hyperparameter that specifies the chunk size. As mentioned in the conclusion, we realize that this way of chunking is quite limiting and future work should study dynamic ways of chunking. Above we have shown how choosing various chunk sizes affects the performance of the model.
> > > > > -  What are the model sizes? Is there any difference when using the recurrent model presented here versus the other transformer models?
> > > > >
> > > > >       In terms of the model sizes, we ensure that in all our experiments the proposed model has similar size as baselines. For example, in the image classification experiments, the proposed model uses 9,649,546 parameters while ViT uses 10,253,578 parameters and Swin Transformer uses 10,438,264 parameters.
> > > > > - How does your method compare to [22], [53] (and other similar papers) in the experiments you presented?
> > > > >
> > > > >   Already presented above. [22] -> Feedback Transformer [53]-> BRIMs
> > > > > - Do you use "bidirectional" (or non-causal mask) for language modeling within each chunk?
> > > > >
> > > > >     No, we use only unidirectional information within the chunk for language modeling

---

> > > > > > ### Author Response · Authors · 2022-08-02
> > > > > > **Regarding Static Chunking Limitation**
> > > > > >
> > > > > > The reviewer has also asked what would happen if critical input data is chunked between two related pieces of information. To test this we perform an experiment in Text Classification Benchmark from LRA in which for each sentence, we introduce spaces into the sentence such that each word is divided into two chunks. Therefore, each chunk sees that 2nd part of the previous word and the 1st part of the next word. We find that in this case the performance of the model drops from 82.08% to 81.29%. The drop is not that significant but this still highlights the limitation of static chunking which future work should address.
> > > > > >
> > > > > > We hope that this rebuttal has addressed the concerns raised by the reviewer. If the reviewer feels that any concerns have not been addressed we would be happy to engage in a discussion. We would also be happy to perform any more experiments that reviewer thinks we should perform.

---

> > > > > > > ### Comment · Reviewer_CN5N · 2022-08-08
> > > > > > > **Rebuttal Response**
> > > > > > >
> > > > > > > Thanks to the authors to take time and reply to my questions. I have modified my score to reflect it.
> > > > > > >
> > > > > > > First, I would like to apologize for the misunderstanding on the comment about SOTA. Thanks for pointing that out.
> > > > > > >
> > > > > > > The claim "In our work, fast module operates on fast time-scale and the TLB module operates on slower time-scale" hasn't been properly shown in an experimental setting. Time-scale refers to the information in the input. For example, "Multi-timescale Representation Learning in LSTM Language Models", Mahto et al., 2021, showed that multiple time-scales exist inherently in an LSTM model, and they can be controlled based on Tallec and Olivier ("Can recurrent neural networks warp time?"). Therefore, I disagree about RIMs being unable to capture both multiple timescales and multiple streams.

---

> > > > > > > > ### Author Response · Authors · 2022-08-08
> > > > > > > > **Thank you**
> > > > > > > >
> > > > > > > > We thank the reviewer for reading our rebuttal, and increasing their score. Thank you.
> > > > > > > >
> > > > > > > > **I disagree about RIMs being unable to capture both multiple timescales and multiple streams.**
> > > > > > > >
> > > > > > > > RIMs is a pretty interesting work. We agree that in RIMs (an ensemble of modules interacting with each other via bottleneck of attention), it's possible that the different modules can implicitly capture multiple time-scales and multiple streams.
> > > > > > > >
> > > > > > > > One can think of characterizing the idea along 3 axis:
> > > > > > > >
> > > > > > > > - multi-stream: Multiple streams with different parameters and operating on "different-kinds of information".
> > > > > > > > - multi-component: Each stream is composed of "multiple" modules.
> > > > > > > > - multi-scale: Multiple streams operate at different time-scales.
> > > > > > > >
> > > > > > > > The proposed architecture is  multi-stream, multi-component and multi-scale. RIMs is multi-component. Previous work [1] has found evidence that **optimizing different modules** at different time-scales can improve the performance of RIMs.
> > > > > > > >
> > > > > > > > In this work, we propose an architecture that has information asymmetry property as an explicit inductive bias. Fast stream has information from local context, and slow stream has information from distant past. Fast stream relies on information from slow stream for relevant contextual information. This "asymmetry" forces the representation to capture relevant contextual information.
> > > > > > > > In TLB, there are certain modules which focusses on low-level perceptual information, and some on high level abstract information while in RIMs all the modules focusses on low level perceptual information.
> > > > > > > >
> > > > > > > > Across different experiments (LRA, language modelling, visual classification, imitation learning, self-supervised learning), we show the usefulness of the proposed method.
> > > > > > > >
> > > > > > > > We would be happy to provide any more clarification which can improve the paper.
> > > > > > > >
> > > > > > > > [1] https://arxiv.org/abs/2105.08710, Fast and Slow Learning of Recurrent Independent Mechanisms
> > > > > > > >
> > > > > > > > Thank you for your time and consideration. We appreciate it.

---

> > > > > > > > > ### Author Response · Authors · 2022-08-09
> > > > > > > > > **Further Feedback: End of discussion period**
> > > > > > > > >
> > > > > > > > > Dear. Reviewer,
> > > > > > > > >
> > > > > > > > > We thank the reviewer for taking time in reading our rebuttal. Since rebuttal period is coming to an end, we want to make sure we can address all the concerns reviewer has. We feel like comments by the reviewer has helped us to improve the presentation of the paper, so it would be useful for us to provide any further clarifications, and accordingly incorporate changes.
> > > > > > > > >
> > > > > > > > > If the reviewer feels we have addressed all their concerns, it would be useful if the reviewer can improve their score/rating.
> > > > > > > > >
> > > > > > > > > We appreciate the reviewer's feedback and time.
> > > > > > > > >
> > > > > > > > > Thank you.

---

> > > > > > > > > ### Comment · Reviewer_CN5N · 2022-08-09
> > > > > > > > > **Thank you for your further comments**
> > > > > > > > >
> > > > > > > > > I have read your last reply and appreciate your quick response.
> > > > > > > > >
> > > > > > > > > I keep my current assessment, updated during the rebuttal phase.

---

### Official Review · Reviewer_EwzR · 2022-07-10

**Rating:** 5
**Confidence:** 5
**Soundness:** 2 fair
**Presentation:** 1 poor
**Contribution:** 2 fair

**Summary:**

This work addresses the aspect of memory (state) and its expressiveness in deep learning models for sequences. Recurrent neural networks (such as LSTMs and GRUs) on one end compress the entire history into a single state representation vector while on the other end the memory state in Transformers is a continuously appended list of (key, value) vectors from all previous timesteps and therefore has a higher fidelity representation of the entire past. They seek to find an optimal balance between compression and expressiveness of these 2 contrasting forms of memories. Their proposed solution strategy is a kind of best-of-both- worlds which has 2 processing streams where the “slow” stream is instantiated as a recurrent module which updates a set of state vectors (once every chunk size steps of the input) using a cross attention mechanism and the “fast” stream which updates its internal representation at every input timestep. This “fast” stream is instantiated as a Transformer Encoder except with the addition of cross-attention blocks as well in addition to the standard self-attention. They empirically evaluate their proposed architecture against several standard Transformer models across various domains such as visual representation learning (ViT), offline reinforcement learning (Decision Transformer) and language modeling (Transformer-XL).


**Questions:**

Following are a list of questions (please note that the order does not represent the importance/significance):

1) Several component modules which form the overall TLB architecture bear resemblance to several other recent works like the Perceiver [1], Slot Attention [2], HCAM [4] (some of which the authors have mentioned). Therefore it would be insightful to understand why the specific model architecture of the TLB architecture is advantageous to these recent architectures that also are motivated by similar considerations such as latent bottlenecks for compression of information using cross-attention mechanisms (Perceiver [1], Slot Attention [2]), representing information at different temporal granularities (HCAM [4] and Hierarchical Transformer [5] in Transformers), a dual stream of information processing for fast and slow learning (DCEM [3]).

2) As one of the ways the authors motivate the use of the TLB architecture seems to be the gains in training data and memory. It would help to show some quantitative metrics (number of training steps to converge, hardware requirements in terms of memory and wall clock times for training the TLB vs the vanilla Transformer) on a few representative benchmarks/tasks to support it.

3) While I greatly appreciate the efforts of the authors to test their proposed TLB architecture on a variety of tasks across different domains. I find the analysis and insight from each of the experimental results lacking depth. The authors do not present an analysis of the results beyond how the model improves evaluation metrics on the various empirical benchmarks. It would be beneficial to move some of the experiments/results to the Appendix and try to show deeper analysis of the empirical performance and reasons behind it on fewer tasks. What is the broader takeaway message/insight for a practitioner beyond the empirical gains it yields to varying degrees over standard Transformer architectures on various tasks?

4) In Figure 3, on the BossLevel the average success rate of the Transformer (in orange) seems to still be rising and not converged yet (looking at the slope of the line) whereas the Transformer + TLB (in blue) has largely flattened out. Could it be that while the Transformer + TLB converges faster it could yield a worse final performance than the standard Transformer (in orange). Could the authors comment if tracking the performance after further timesteps the standard Transformer would yield a higher success rate?

5) On some benchmarks the authors show the average and standard returns over 10 seeds whereas on others only on 3 seeds (Tables 1 & 2). Could the authors please use a consistent experimental protocol throughout (atleast 5 seeds consistently)? Also, the authors report evaluation metrics on visual datasets inconsistently i.e. in Tables with classification accuracy (Tables 1 & 2) and some others as bar plots (Figure 2). Could the authors maintain consistency here as well? Could the authors clarify how many seeds are used in the self-supervised learning experiment (Figure 2)?

6) Lines 224-227: On the similarities of the proposed model with concurrent work on the Block Recurrent Transformer. The authors claim that they have found performing cross attention and self attention in sequential performs better than doing them in parallel as used by the Block Recurrent Transformer (lines 224-225). I could not find any experiment in the paper where they provide empirical evidence to support this claim. Further, they also state that “they use special tricks to deal with some instabilities in their case”. It is unclear what the authors are exactly referring to here, could the authors please elaborate on what these “special tricks” are and what the potential downsides of them are as compared to the proposed TLB architecture?

Overall, I like the high-level idea and greatly appreciate the efforts of the authors to empirically test their proposed architecture’s performance on a wide variety of domains. But the presentation (writing) of ideas and motivation behind the specific design choices seem rather unclear to me. Also, the design of experiments do not help to get a deeper insight about its working and also why this specific architecture is better than several previous related ideas to improve the standard Transformer architecture. I think it would help gain a better understanding of the proposed architecture and its several parts by designing careful experiments that show why these specific choices are better than some competing alternatives that attempt to do the same as highlighted above.

References:

[1] Perceiver: General Perception with Iterative Attention, Jaegle et. al, 2021.

[2] Object-centric Learning with Slot Attention, Locatello et. al, 2020.

[3] Grounded Language Learning Fast and Slow, Hill et. al, 2020.

[4] Towards mental time travel: a hierarchical memory for reinforcement learning agents, Lampinen et. al, 2021.

[5] Hierarchical Transformers are more efficient language models, Nawrot et. al, 2021.

[6] Ordered Memory, Shen et. al, 2019.


************************** Post-Rebuttal Update  ******************************************************

I've updated my rating to a 5 post-rebuttal in light of the changes/ improvements
made to the paper. I believe this submission is very much a borderline one and its
overall quality could greatly be improved with more revisions that demand more time
beyond the rebuttal period. I hope the authors can take the time to address any
remaining concerns raised all the reviewers well to continue to improve the submission.

***************************************************************************************************************

**Limitations:**

The authors only very briefly describe the limitations of their method by simply commenting about the fixed chunk size used, a hyperparameter which requires some domain knowledge to tune. Ideally, it would be great if the authors can show at least one task/domain on which the proposed TLB architecture struggles with and comment on the plausible reasons behind it. This would  benefit the research community to develop a deeper understanding of the proposed architecture including its limitations and also allow for future work to build on these insights.


**Strengths And Weaknesses:**

Strengths:

- The high-level idea behind the proposed TLB architecture is interesting and the authors have taken a lot of effort to show empirical results of their proposed architectural modification to the standard Transformer on a wide range of domains such as offline reinforcement learning, visual representation learning, language modeling etc. and their benefits.

- The proposed architecture shows very good empirical gains on Atari in the offline setting over the Decision Transformer, and to a lesser extent on classification tasks on CIFAR-10 and CIFAR-100 length (i.e. image resolution) generalization, language modeling on enwiki8 dataset and learning long-range dependencies (ListOps and Text Classification) tasks. The proposed TLB architecture provides benefits both in terms of computational efficiency and performance.


Weaknesses:

- Please cite related work on fast and slow learning mechanisms such as the DCEM model [3] which uses a dual stream processing to allow for fast associative memories and slow episodic memories in object-word association/RL tasks, Hierarchical Transformer [5] which biases the Transformer to process long sequences in a more efficient manner by using inductive biases to model the sequence hierarchically, Ordered Memory [6] which learns to represent a hierarchical structure in RNNs.

- The novelty of the overall architecture seems marginal and that it is more an amalgamation of several components introduced by other work such as Perceiver [1], Slot Attention [2], DCEM [3], HCAM [4], Hierarchical Transformer [5] which have similar motivations behind their architectural improvements to the standard Transformer model. Regardless, I think there could be significant value in this specific architecture. But the current experimental design and analysis of results does not attempt to provide an answer to this. It is difficult for me to judge what the main takeaway messages are in terms of main design principles and how much each of the modules (design choices for sub-components) contribute towards the varying amounts of empirical gains observed on the different tasks.

- The presentation of technical contributions and their motivations, design of experiments and analysis of results can be improved. (see below)


Comments on writing, presentation and typos:

- In general, I found the flow of argumentation in the Introduction section very winding and confusing. At certain parts it almost reads like a historical account of research in deep learning for sequence processing. Currently the motivation for the central research problem is expressed in a scattered manner throughout the section but not explicitly stated in a succinct manner. It would greatly help if the authors introduced and motivated their research question in a succinct manner and how their proposed solution directly addresses this challenge. The writing in the introduction can be significantly modified to favor brevity and precision about the research question being addressed and its significance.

- In Section 2, again the flow of argumentation is very winding and confusing. It would help to directly get to the main points (section 2.1 can be cut down significantly to deliver the main points). Further, in section 2.2 the paragraph on preliminaries can be significantly cut down. At this point, the multi-head attention module and Transformer architecture are fairly standard in the deep learning toolkit that they can be simply referenced. Only specific modifications applied to these standard modules used or the way they’re combined in the overall TLB model architecture can be explained.

- Further, the mathematical notation used in section 2.2 is unclear. I’m confused as to what the intended notational format is with regards to usage of capital letters, small letters and boldface characters? Is it that capital letters indicate a set, small letters a and boldface a vector? If so, then there are several inconsistent usage of this notation throughout section 2.2 which reduces the clarity and readability of equations etc. I request the authors to specify the notation format used upfront and be consistent with it.

- The authors seem to only present the details of the proposed approach . I believe it is absolutely essential that this be present (at least large parts of it) in the main text as it is the central technical contribution of the paper. Space can be created by re-structuring/condensing other sections of the paper as mentioned earlier.

- Lines 34-35: “... which force the entire hidden state to be compressed into a single hidden state” -> “... entire history into a single hidden state”?

- Lines 40-41: “…, especially in the presence of long-range dependencies” -> missing citation. Also sentence construction can be improved -> use of long-sequences & long-range dependencies is redundant.

- Lines 53-54: “... and may bias them towards modeling unnecessary, irrelevant or noisy interactions” -> unclear what this means and how it is connected to motivating the TLB?

- Line 105: “... is not completely new.” -> “is not a completely new idea”?

- Line 104: “.... and slow processing mechanisms in machine learning” -> “... and slow processing mechanisms in neural networks/deep learning” a more precise phrasing perhaps?

- Line 115/215: “… [53] explored” and Line 118: “... [22] showed”, please correct the citation format appropriately to XYZ et. al [53].

- Line 126: “We first describe our notation” -> uninformative sentence - can be removed.

- Repeated usage of the phrase “temporal latent bottleneck” in sections 1 & 2 can be replaced with the abbreviation TLB which is later adopted in section 4.

- Line 158: “As mentioned previously,... ” -> uninformative/redundant phrase, can be removed.

- Line 324: “Baby AI. BabyAI [15]” -> minor typo.

- Easily noticeable that the spacing between sections and subsections has been significantly altered (reduced) with respect to the standard template (for instance spacing before sections 2, 3, 4). Please adhere to the conference template.

---

> ### Author Response · Authors · 2022-08-02
> **On novelty of the proposed approach**
>
> We would like to thank the reviewer for the detailed comments. We appreciate that the reviewer likes our proposed idea. We are grateful to the reviewer's time and feedback. Reviewer's feedback has helped us to improve the exposition of the paper.
>
> We would first like to give a general update that we were able to improve the performance of the proposed model on the text classification task from the long range arena benchmark. We now achieve a performance of 82.08% as compared to 67.8% that we reported in the submission. This beats the highly competitive S4 model [5] which achieves 76.02%. The main improvement comes from reducing chunk size to 10 (+6%) and adding local positional embedding within each chunk (+8%).
>
> **The novelty of the overall architecture seems marginal and that it is more an amalgamation of several components introduced by other work such as Perceiver [1], Slot Attention [2], DCEM [3], HCAM [4], Hierarchical Transformer [5] which have similar motivations behind their architectural improvements to the standard Transformer model**
>
> Here's a description of how these methods differ from the proposed method.
>
> The original idea of having a set of slots, and each slot attending to a distinct region of the input using cross-attention was initially proposed in RIMs [6]. Slot Attention extends the idea for learning representation of slots from **static** scenes using iterative cross-attention (so the only connection to our work is the use of slots which is not the novelty of slot-attention). Perceivers also use cross-attention to map a full input array into a smaller latent array and perform all subsequent attention operations in the resulting latent space.  However, because each model latent attends to all inputs regardless of position, Perceivers cannot be used directly for autoregressive generation, which requires that each model output depend only on inputs that precede it in sequence. Both slot-attention and Perceivers don't have (a) multiple streams of computation intreacting with each other via bottleneck of attention (i.e., top-down attention); (b) information asymeetry between fast stream and TLB module i.e. fast stream operates within a chunk and TLB modules aggregates information across chunk.
>
> DCEM [3] is an interesting work. The way DCEM uses the idea of dual coding is reminicant of ideas from Memmory networks (or relational memory networks). The agent parameterized as LSTM writes summary of linguistic representations to the "keys" of the memory, and the vision representation to the values of the memory. The agent computes a query which is matched to the keys in the memory, and the result of the procedure is usd by the agent to update it's internal state. It's an "interesting" blend of ideas from attention and memory networks. DCEM can be seen as a "efficient" version of the Trasnformer agent also mentioned in the original DCEM paper (see Section 4 in experiments section DCEM v/s Transformer-XL) "These results suggest that DCEM
> is more ‘working-memory-efficient’ than the Transformer agent"). Important difference from our work is the presence of information asymmetry and both the streams operating at different time-scales. In our work, TLB is parameterized as a set of slots, and have internal dynamics which gets updated using information from the fast stream. In DCEM, the memory does not have recurrent dynamics. So, in short DCEM can be seen as much improved version of Transformer-XL for the multi-modality tasks.
>
> HCAM [4] is also an interesting work. HCAM is a "transformer" version of the Sparse attentinve backtracking paper, where the authors have shown that sparse, top-k retrieval of memory chunks can improve credit assignment in LSTMS with attention (mentioned in the introduction of HCAM paper) "(Relatedly, Ke et al. [25] have shown that sparse, top-k retrieval of memory chunks can improve credit assignment in LSTMs with attention.)". Both SAB, HCAM and other works in Hierarchical RNNs/Transformers don't have information asymmetry and hence don't have bias towards "compressing" representions as present in the proposed work. We show that the compression as a result of information asymmetry between fast module and TLB module improves generalization and adaptation. In the proposed model, the fast stream ONLY has access to information from the current stream, whereas TLB module has access to coarse level information. The only way for the fast stream to access information about the distant past is to use information from TLB. Such an information asymeetry has shown to improve generalization performance of trained policies in RL. In this work, we propose an architecture which explicitly has an information asymmetry constraint. Both HCAM and SAB relies explicitly on recalling ifnormation from the distant past, whereas we show that TLB module compresses information relevant to the fast stream.
>
> [5] Efficiently Modeling Long Sequences with Structured State Spaces Gu et al 2021
>
> [6] RIMs

---

> > ### Author Response · Authors · 2022-08-02
> > **On Novelty and More Analysis**
> >
> > Other than the above mentioned points, "empirically" another important difference is we show the utility of the proposed architecture in various different problems ranging from language modelling, self-supervised learning, imitation learning, perceptual vision problems. All the above mentioned papers explored the architecture for a particular problem i.e. slot attention for object centric learning, HCAM for distant credit assignment in RL, Perceivers for non-autoregressive tasks, DCEM for multi-modality processing.
> >
> > In this rebuttal, we provide a deeper analysis of our model and try to address the questions raised by the reviewer.
> >
> > **Regarding more analysis and insight**
> >
> > We have conducted more experiments analysing the proposed model and its various components. We have updated the main paper with these experimental results. Here, we provide a brief overview of our analysis experiments -
> >
> > *Impact of Top-Down Information*
> >
> > The TLB is not only important because it provides information from the past but also because it provides high-level information through top-down communication. We design a baseline which is similar to our proposed Transformer + TLB model but it has a different TLB module after each layer as opposed to having only one TLB module after the last layer. Thus, each layer L of the perceptual module reads and writes information from the TLB module specific to layer L. Therefore, in this case the TLB module only provides information about the past, it does not contain any top-down information since the TLB module at layer L does not communicate with the lower layers of the perceptual module. We evaluate this baseline on CIFAR10 image classification, IMDB text classification, and ListOps. The results are as follows:
> >
> > - CIFAR10 Image Classification - ViT + TLB: 94.79%, ViT + TLB (No Top-Down): 93.75%
> > - ListOps: Transformer + TLB: 38.2%; Transformer + TLB (No Top-Down): 37.57%
> > - IMDB Text Classification - Transformer + TLB: 82.08%; Transformer + TLB (No Top-Down): 80.68%
> >
> > This shows that the TLBs ability to provide top-down information along with information from the past is important for the model to perform well. Only providing summarized information from the past is not enough.
> >
> > *TLB Captures the most important information required to solve the downstream task*
> >
> > Our hypothesis here is that the limited capacity of the TLB allows it to focus on the most important information in input required to solve the downstream task ignoring everything else that is not relevant. To confirm our hypothesis we perform two experiments:
> >
> > - First, for the image classification experiment, we visualize the patches that the TLB pays most attention to. To do this we first store the attention scores that the TLB pays to each patch in the image when the perceptual module is writing information to it (i.e. attention scores computed in the last step of Algorithm 1 in the Appendix). We only keep those patches in the image that fall in the top 30% of the attention scores masking everything else. We present the result of this visualization experiment in Figure 2 in the main paper. We can see that the TLB pays most attention to foreground of the image which shows that the TLB focuses on the most important information in the image learning to ignore irrelevant information.
> > - Second, we evaluate the proposed Transformer + TLB model on the copying task used in [1]. In this task the model is first given a sequence of 10 digits as input and it is supposed to output the same sequence after an interval of blank inputs. We describe the task in detail in Section 3.4 of the paper. The length of the blank inputs in the middle can be as high as needed resulting in very long sequence lengths. For a model to do well, it needs to remember the original sequence across long time-scales. We compare the proposed Transformer + TLB model to the Feedback Transformer [2] which has been shown to work well on this task. We compare both the models for varying sequences lengths from 100 to 600 as shown in Figure 5 in the main paper. We see that for higher sequence lengths the performance of the Feedback Transformer starts dropping while the proposed model still achieves perfect accuracy. This shows that the TLB retains the original sequence across long time scales without overwriting any of its information in the presence of blank inputs. The Feedback Transformer is unable to do so.
> >
> > These two experiments confirm that the TLB learns to capture and retain important information without being affected by noisy or uninformative inputs.
> >
> > [1] Long Short-Term Memory Hochreiter et al 1997
> >
> > [2] Addressing Some Limitations of Transformers with Feedback Memory Fan et al 2021

---

> > > ### Author Response · Authors · 2022-08-02
> > > **More Analysis of The proposed model (contd)**
> > >
> > > *TLB retains information across long time-scales*
> > >
> > > One of our main hypothesis is that TLB acts a long-term memory that stores information across long time scales. We already confirm this hypothesis by showing strong performance gains on task that require long-term contexts such as IMDB byte-level text classification (sequence lengths of ~4k), ListOps (sequence lengths of ~1k), and copying (sequence lengths of upto 600). To further give more concrete proof of this hypothese we design a baseline in which each chunk attends to only past few chunks instead of attending to the TLB. If this baseline achieves almost similar performance to TLB, this would show that the TLB only contains information from the recent few chunks. We evaluate this baseline on the ListOps task and find that it achieves an accuracy of 32.10% which is much lower than that of the proposed model Tranformer + TLB model which achieves 38.2%. This shows that the TLB caputures longer term information than the past few chunks.
> > >
> > >
> > > *Effect of the chunk size and number of TLB state vectors*
> > >
> > > The chunk size and number of TLB state vectors are hyperparameters. We vary the value these hyperparameters and study their effect on performance of the model.
> > >
> > > - **Chunk Size**: We try the following values for chunk size on the ListOps task: 2, 20, 100, 500, 1000. We present the results for this analysis in Appendix Table 6. We find that the model achieves best performance at chunk size 100, above or below which the performance drops.  Lower chunk sizes can potentially lead to a lot of information to integrate across chunks which might make the TLB forget important information more quickly and higher chunk sizes can lead to too much information to write in one chunk which can also lead to unwanted forgetting. We also evaluate various chunk sizes for the multi-task BabyAI experiment (Appendix Figure 9) and reach a similar conclusion that there is an optimal chunk size above or below which the performance drops.
> > > - **Number of State Vectors**: We try the following values for the number of state vectors on the ListOps task: 1, 10, 20, 200. Here also we find that there is an optimal number of state vectors above or below which the performance drops. Less number of state vectors can lead to very low capacity in the Temporal Latent Bottleneck leading to loss of important information. Similarly, a high number of state vectors can lead to a high capacity in the Temporal Latent Bottleneck leading to it capturing a lot of unnecessary of noisy information.
> > >
> > > *Consequences of Static Chunking*
> > >
> > > One limitation of our model is that we have to specify a static chunk size. In some cases, this may result in a single piece of critical information being dividing across multiple chunks. This may lead to a drop in performance. To test how much of a drop can we expect in this case, we perform an experiment in the Text Classification Benchmark from LRA. For each sentence, we introduce spaces into the sentence such that each word is divided into two chunks. Therefore, each chunk sees that 2nd part of the previous word and the 1st part of the next word. We find that in this case the performance of the model drops from 82.08% to 81.29%. The drop is not that significant which shows that even if critical information is divided across two chunks, the model should suffer only very minor drop in performance. We believe that static chunking is a limitation of our model and future works should introduce dynamic ways of chunking.

---

> > > > ### Author Response · Authors · 2022-08-02
> > > > **On Sample Efficiency and Compute Efficiency**
> > > >
> > > > **On sample efficiency and Number of Training Steps**
> > > >
> > > > The reviewer has asked for quantitative metrics evaluating the sample effeciency of the model. Previously, we had experiments showing that proposed Transformer + TLB model converges faster than the baseline Transformer on the Baby AI benchmark. In the rebuttal, we show more results showing the sample effeciency of the proposed model:
> > > >
> > > > - Sample Efficiency in ListOps: We evaluate the sample efficiency of the proposed Transformer + TLB model and the baseline Transformer LS model on the ListOps dataset. For this, we train both the models for one pass over the entire ListOps dataset i.e. no example is seen more than once by the model. We plot the convergence curves against the number of samples in Appendix Figure 7 (a). We can see that the Transformer + TLB model is much more sample efficient than the Transformer LS baseline converging in almost half as many examples as the baselines.
> > > > - Sample Efficiency in Copying: We evaluate the sample efficiency for the copying task in similar manner as for the ListOps task. Both the proposed model and the baseline Feedback Transformer see only unique examples during training. We calculate the number of samples required for both the models to achieve perfect accuracy.
> > > >     | Sequence Length | Feedback Transformer | Transformer + TLB |
> > > >     | --------------- | -------------------- | ----------------- |
> > > >     | 100             | 11800                | 6200              |
> > > >     |200 | 16600 | 9100 |
> > > >     | 300 | 35100 | 12700 |
> > > >     |400 | NA | 14600 |
> > > >     | 500 | NA | 13600 |
> > > >     |600 | NA | 19300 |
> > > >
> > > >     We can see that in all cases Transformer + TLB needs much lesser samples than the baseline to converge to perfect accuracy. NA in the above table indicates that the model does not converge to perfect accuracy.
> > > >
> > > > **On memory and wall-clock time requirements**
> > > >
> > > > We evaluate the memory and wall-clock times for the proposed model w.r.t to the baseline Transformer model on the text classification task from the LRA benchmark (sequence length of around ~4k). We present the results in the following table. In the following table x indicates the value of the corresponding metric for a vanilla transformer.
> > > >
> > > > | Chunk Size | 1000 | 100 | 40 | 20 | 10 |
> > > > | ---------- | ---- | --- | -- | -- | -- |
> > > > | Inference Speed |  3.5x | 3.6x | 3.3x | 2.2x | 1.2x |
> > > > |    Inference Memory |  0.09x | 0.08x | 0.12x | 0.08x | 0.1x |
> > > > |    Training Speed |  4.4x | 4.4x | 2.2x | 1.4x | 0.7x |
> > > > |    Training Memory |  0.14x | 0.08x | 0.49x | 0.40x | 0.42x |
> > > >
> > > > Therefore, a Transformer + TLB model with chunk size 20 has 2.2x faster inference than a Transformer, uses 0.08x inference memory of a Tranformer, trains 1.4x times faster than a Transformer, and uses 0.40x memory of a Transformer during training. We refer the reviewer to Appendix section 7.3 (last para) for more details on this.
> > > >
> > > > **On BossLevel**
> > > >
> > > > The reviewer says that in the BabyAI BossLevel results from Figure 4, it seems that the baseline has not converged yet. We agree with this point. Therefore we run the BossLevel experiment again for 100k steps (the one reported in the paper was for 50k steps). We have updated Figure 4 paper with the new results. From the new results, it seems that both Transformer + TLB and the baseline converge to a similar score.
> > > >
> > > > **On discrepency in the number of seeds and reporting of experiments**
> > > >
> > > > We apologise for the discrepency in the number of seeds in different experiments. We ran 10 seeds for the Atari experiments since the variance was high in those experiments hence we ran the most number of seeds for that experiment. For the other experiments, we don't see much variance in the seeds. We do agree that we should report atleast 5 seeds for each experiment. Given the limited time of 1 week, we could not do this for rebuttal but we will run 5 seeds for all experiments soon. For all the new experiments that we report in the rebuttal, we have run 5 seeds.

---

> > > > > ### Author Response · Authors · 2022-08-02
> > > > > **On Comparison to Block Recurrent Transformer and Writing Improvements**
> > > > >
> > > > > **On comparison with Block Recurrent Transformer [4]**
> > > > >
> > > > > In the submission we claimed that sequentially applying cross attention and self attention works better as compared to doing them in parallel as done in Block Recurrent Transformer. The reviewer says that we did not provide any experiment for this claim. We apologise for ommiting this result from the main paper. We present those results here for the CIFAR10 dataset.
> > > > >
> > > > > | Model | 64 x 64 | 128 x 128 |
> > > > > | ----- | ------- | --------- |
> > > > > | Sequential (Ours) | 94.79 | 84.38 |
> > > > > | Parallel (Block Recurrent Transformer) | 85.42 | 66.93 |
> > > > >
> > > > > Regarding special tricks used in Block Recurrent Transformer. In section 3.5 of the Block Recurrent Transformer paper [https://arxiv.org/abs/2203.07852], the authors say that "the training stability is quite sensitive to how the gates are initialized". To mitigate this issue they initialize the bias to small non-zero values. Also, they "add a constant of -1 and +1 to the input and forget gates (see Eq. 4) to initially bias the gate to “remember”". These are the special tricks that we refer too. Our proposed model does not need any such tricks for performing well and we do not observe any training stability issues.
> > > > >
> > > > > Another difference between Block Recurrent Transformers and our paper is that they mainly focus on Natural Language Processing tasks while we focus on a much broader variety of tasks.
> > > > >
> > > > > **Regarding Suggestions on Writing**
> > > > >
> > > > > We thank the reviewer for detailed suggestions on our writing. We have updated the introduction to be more clear about the problem that we study. We have also cut down on section 2.1 and removed the preliminaries. We have also tried to simplify the notation to be more clearer to read. We only use small letters to denote individual chunks in a sequence, everything else is denoted as capital letters. We have also added all the new experiments we performed for the rebuttal to the paper. We hope that the overall writing is more clear now. We would be happy to incorporate any more suggestions that the reviewer may have.
> > > > >
> > > > >
> > > > > We hope that this rebuttal has addressed the concerns raised by the reviewer. If the reviewer feels that any concerns have not been addressed we would be happy to engage in a discussion. We would also be happy to perform any more experiments that reviewer thinks we should perform.
> > > > >
> > > > > [4] Block Recurrent Transformers Hutchins et al 2022

---

> ### Author Response · Authors · 2022-08-07
> **Follow up on the rebuttal**
>
> Dear. Reviewer,
>
> Thanks again for taking the time to review our paper and providing feedback.
> We have updated the introduction of the paper, as well as added detailed replies to all of the reviewer's comments (after conducting more experiments). We felt that the reviewer's review was very helpful and instructive, so it would be very helpful to get feedback on our changes (and also update rating if applicable). We would also be grateful if the reviewer could let us know if there are any other concerns that we can address further.
>
> Thanks for your help and time. :-)

---

> ### Comment · Reviewer_EwzR · 2022-08-08
> **Response to the rebuttal updates - (1/2)**
>
> I’m really sorry for my slow response to the discussion. Firstly, I’d like to appreciate the author's efforts to run additional experiments/analysis and provide detailed responses to address the concerns raised. Following are some comments in light of the changes made during the rebuttal:
>
> - While the authors have provided additional analysis on the computational efficiency as I suggested. These empirical results seem to be relegated almost entirely to the Appendix. Since one of the key advantages (and even motivation) of the proposed TLB model is its computational benefits over the standard Transformer. It then seems necessary that the authors present these computational efficiency scores (at least the main one) in the main text since they back-up the claims about computational efficiency and associated upper bounds (big O) calculated in the Introduction section with some real-world metrics. The sample-efficiency gains compared to the Feedback Transformer reported are encouraging.
>
> - The Introduction has improved and seems to be rather succinct and directly addresses the research question of interest.
>
> - I was not able to gather the main takeaway from looking at Figure 2? What are the 4 panels showing? All that I can seem to parse is that there are many patches being masked and few attended to (predominantly the patches corresponding to an object in the center) but I can’t infer much beyond that. The associated text (and caption) is also not helpful. Please improve the visualization further and use the associated text to provide context information and direct the readers’ focus to the relevant aspect of the visualization. Further, what if we modulate that top-x% (x=[1, 5, 10, 25])? How do the patches attended to vary? For x=1 or 5, intuitively it should only focus on the patches that are most relevant to ascertain the object class information. And as we increase x, the model progressively increases the “field” of vision to include lesser informative patches used to determine the class label. Could the authors check if this is the behavior that’s being observed empirically? Further, can the authors provide more such visualization on a few other downstream tasks as well (can be added to the Appendix)?
>
> - The No top-down conditioning ablation in Table 2 seems to only very slightly reduce model performance (94.79 -> 93.75 (less than 1%) and 84.38 -> 83.59 (again less than 1%). Since no standard deviations have been reported for the results in Tables 1 & 2 it is difficult to really judge whether the TLB does better than the baselines. Further, as I mentioned before it is helpful to at least report mean and std. deviation over 5 seeds at the minimum for all experiments. Since these are experiments on CIFAR datasets I expect the compute budget needed for additional 2 seeds should be very reasonable. It is possible that this marginal performance gap could be reduced when evaluated over a greater number of seeds. It is helpful to reject this possibility.
>
> - I will give the benefit-of-doubt to the newly added (missing in the initial submission) results comparing TLB with the Block Recurrent Transformer on image classification on the CIFAR10 dataset. Does the Block Recurrent baseline shown only differ wrt the application of cross-attention with all the modeling choices such as hidden size, number of attention heads, attention head size etc. kept the same as TLB in this experiment? Basically, have the authors controlled for all the other possible knobs that affect the performance of the models and isolated simply the design strategy of how the cross-attention is applied in this experiment? Have they tuned the training strategies for the Block Recurrent Transformer as well? Could they please elaborate on this in the Appendix.
>
> - Why is the core contribution of the paper and its technical details described in Algorithm 1 being relegated to the Appendix? I think the core contribution described in Algorithm 1 with most of its key details definitely needs to be included in the main text.
>
> - There have been improvements made to the Perceiver model published recently such as Perceiver-AR [1] at ICML 2022 which allow the Perceiver to also perform autoregressive modeling by using causal masking. These recent developments need to be discussed and differences highlighted.
>
> - There have been improvements made to Slot Attention as well, such as SAVi [2] which allows for the set of slots to maintain state and remain stable over a full video sequence. Further, the slots in this model can be conditioned using high-level contextual information such as center-of-mass, position, segmentation masks etc. to represent specific scene components. These recent developments need to be discussed and differences highlighted.
>
> --- end of part 1/2 ---

---

> > ### Comment · Reviewer_EwzR · 2022-08-08
> > **Response to the rebuttal updates - (2/2)**
> >
> > - Regarding my comment about the fact that the paper doesn’t provide sufficient empirical insight into why the specific architectural additions proposed are preferable over several related methods that seek to do similar things such as hierarchical modeling of sequences with Transformers, dual-stream (fast and slow) processing in sequence models and use of top-down conditioning.
> > For instance in experiments on the BabyAI or Atari domains (shown in Figure 4) the baseline Transformer doesn’t contain any such additional inductive biases to incorporate hierarchical structure, dual stream of memory and top-down conditioning. So it is difficult to assess these additions as the baselines don’t offer a like-for-like comparison in this sense.
> >
> > - Further, I’m happy that the authors re-ran the experiment for more training iterations to report that the baseline Transformer and the TLB variant on longer training seem to converge to similar scores. This begs the question, why do the proposed architectural additions of chunking not seem to offer any benefit in this domain? Could the authors delve into this more? The authors claim (lines 65-66) that “... the proposed model in a number of domains showing that it consistently outperforms competent baselines showing improved generalization to scenarios not seen during training.” This claim is unsubstantiated as-is and needs to be tempered significantly to better reflect these results on the BabyAI domain and only competitive performance wrt Swin-v2 (Table 1).
> >
> > - So, while the proposed TLB augmented Transformer variations do show varying degrees of performance improvements over the vanilla Transformer, these results do not help to evaluate how these particular proposals for incorporating hierarchical structure based on chunking and top-down conditioning are better or more suitable than competing alternatives that seek to improve the standard Transformer along similar lines. The exception to this is in the case of when Swin-v2 is used as the baseline Transformer model for image classification where Swin-v2 has an inductive bias for modeling hierarchy.
> >
> > - Regarding the inconsistency about using tables and/or bar plots to report, they continue to persist. Tables 1 & 2 report classification results of TLB-based ViT and relevant baselines on CIFAR10/100 whereas in Figure 3, bar plots are used to report classification scores for the SiT model.
> >
> > In general, I feel while the paper is improving, there still remain several aspects relating to the presentation, experimental design/analysis that seem rushed and could do with more refinement that would need a greater time investment. I greatly appreciate the authors’ efforts to incorporate changes to address my concerns. But I feel since there are still some key aspects that need to be addressed/improved I will maintain my original rating of 4 for the paper.
> >
> > [1] General-purpose, long-context autoregressive modeling with Perceiver-AR, Hawthorne et. al, ICML 2022.
> > [2] Conditional object-centric learning from video, Kipf et. al 2021.

---

> > > ### Author Response · Authors · 2022-08-08
> > > **Regarding Presentation and More analysis**
> > >
> > > We thank the reviewer for taking time to read our rebuttal. We appreciate it.
> > >
> > > **These empirical results seem to be relegated almost entirely to the Appendix.**
> > >
> > > We apologize for this. We tried to add as much analysis as possible to the main paper (Figure 2, page 5 last para, page 6 first para, page 8 last para and page 9). Due to lack of space, we had to move some analysis to the appendix. We would shift the analysis on compute benefits to the main paper if accepted.
> > >
> > > **On top-down ablation**
> > >
> > > The reviewer says that we have not reported standard deviation for the experiments performed for showing the effect of top down communication. The results we reported for cifar10 are averaged over 3 seeds. Here are the results with standard deviation -
> > >
> > > CIFAR10 - ViT + TLB: 94.79 +/- 0.32; ViT + TLB (No Top Down): 93.75 +/- 0.24
> > >
> > > ListOps - Transformer + TLB: 38.2 +/- 0.0001; Transformer + TLB (No Top Down): 37.57 +/- 0.003
> > >
> > > Text Classification - Transformer + TLB: 82.08 +/- 0.44; Transformer + TLB (No Top Down): 80.68 +/- 0.34
> > >
> > > We could not run over 5 seeds during the rebuttal phase due to the limited time (and compute since we already ran a large number of experiments), but we will do so after the rebuttal phase. **Note that the Text Classification results have been run over 5 seeds.**
> > >
> > >
> > > **On Figure 2 (cifar10 visualization)**
> > >
> > > In Figure 2, the top row shows the two original cifar10 images while the bottom row shows the same image in which those patches have been masked out where the TLB pays least attention to. In Figure 2, we retain only those patches that are in the top-30% of the attention scores. The main takeaway from this figure is that the TLB learns to pay most attention to the foreground image in cifar10 which shows that it learns the most relevant information required to solve the downstream task. The reviewer has requested to show the same ablation but for various top-k. We vary k from 20 to 50. We have shown the result of this ablation in Appendix Figure 6. We can see that as we increase k, the models field of vision increases to include less informative patches, as predicted by the reviewer.
> > >
> > > **Standard Deviations for Table 1 and Table 2**
> > >
> > > We apologise for omitting the standard deviation from Tables 1 and 2. We did that as the table was taking too much (horizontal) space. Below are both the tables with standard deviations. We will update this in the main paper too.
> > >
> > >
> > > Table 1
> > >
> > > | Model | CIFAR10 | CIFAR10 | CIFAR100 | CIFAR100 |
> > > | ----- | ------- | ------- | -------- | -------- |
> > > | | 64 x 64 | 128 x 128 | 64 x 64 | 128 x 128 |
> > > | ViT | 93.75 +/- 0.21 | 73.18 +/- 0.25 | 69.53 +/- 0.55 | 47.4 +/- 0.43 |
> > > | SwinV2 | 97.66 +/- 0.13 | 84.9 +/- 0.47 |  79.95 +/- 0.36 | 58.59 +/- 0.27 |
> > > | ViT + TLB | 94.79 +/- 0.32 | 84.38 +/- 0.42 | 79.17 +/- 0.13 | 59.19 +/- 0.38 |
> > >
> > >
> > >
> > >
> > > Table 2
> > >
> > > | Model | CIFAR10 | CIFAR10 |
> > > | ----- | ------- | ------- |
> > > | | 64 x 64 | 128 x 128 |
> > > | ViT + TLB | 94.79 +/- 0.32 | 84.38 +/- 0.42  |
> > > | No Past Info | 91.30 +/- 0.24  | 72.92 +/- 0.35 |
> > > | No Top Down | 93.75 +/- 0.24 | 83.59 +/- 0.27 |
> > >
> > > **On Block Recurrent Transformer**
> > >
> > > We have ensured that all the knobs and hyperparameters between the block recurrent transformer and the proposed TLB model are same and they have the exact same number of parameters. There is only difference -
> > >
> > > - We do sequential self and cross attention for the proposed TLB model, while for block recurrent transformer, we do parallel cross and self attention.
> > >
> > > We would like to point that block recurrent transformer is work done in parallel to our work. While their main focus is natural language tasks, we focus on a broader variety of tasks. We will add more details on this experiment in the appendix.
> > >
> > > **Why is the core contribution of the paper and its technical details described in Algorithm 1 being relegated to the Appendix?.**
> > >
> > > We will shift Algorithm 1 to the main paper after the discussion period.

---

> > > > ### Author Response · Authors · 2022-08-08
> > > > **Regarding Baselines**
> > > >
> > > > **Regarding Representative Baselines**
> > > >
> > > > The main components of our architecture are -
> > > >
> > > > - Hierarchical Processing
> > > > - Information assymetry
> > > > - Top-Down information
> > > >
> > > > We compare against baselines the have these components -
> > > > - **Hierarchical Baseline**: As the reviewer mentioned SwinV2 has hierarchical processing and, to our best knowledge, is also the state of the art model for image size extrapolation.
> > > > - **Information Assymetry**: Transformer LS model used as baseline in the long range arena has seperate attention mechanisms to attend to long range and short range information. This results in multiple attention stream - one specialized to coarse grained distant features and one specialized to fine-grained local features resulting in an assymetry of information.
> > > > - **Top-Down Information**: The feedback transformer is a transformer based baseline which also has top-down informatio but does not have multiple scales of processing. We have shown that we outperform feedback transformer on the copying task (section 3.3 in main paper).
> > > >
> > > > We also note that we achieve state of the art results on the text classification task from the long range arena benchmark. We now achieve a performance of 82.08% as compared to 67.8% that we reported in the submission. This beats the highly competitive S4 model  (and many other transformer variants) which achieves 76.02%. The main improvement comes from reducing chunk size to 10 (+6%) and adding local positional embedding within each chunk (+8%).
> > > >
> > > > **Regarding baselines used for babyai and atari**
> > > >
> > > > We use the offline RL setting for the BabyAI and Atari experiments. Most recent works in offline RL such as Decision Transformer [1], Trajectory Transformer [2], GATO [3], and many other works mainly contribute different formulations to frame the reinforcement learning problem as a sequnce modeling problem but the core architecture that they use is still a vanilla transformer. Therefore, we believe vanilla transformer (i.e., Decision Transformer) is a competitive baseline for offline sequential decision making problems.
> > > >
> > > > [1] Decision Transformer: Reinforcement Learning via Sequence Modeling Chen et al 2021
> > > > [2] Reinforcement Learning as One Big Sequence Modeling Problem Janner et al 2021
> > > > [3] A Generalist Agent Reed et al 2022
> > > >
> > > >
> > > > **Regarding the inconsistency about using tables and/or bar plots to report, they continue to persist. Tables 1 & 2 report classification results of TLB-based ViT and relevant baselines on CIFAR10/100 whereas in Figure 3, bar plots are used to report classification scores for the SiT model.**
> > > >
> > > > We apologise for this and we will change the sit results to a table in the main paper. During rebuttal times, we focussed on most important ones (like changing introduction, and running experiments).

---

> > > > > ### Author Response · Authors · 2022-08-08
> > > > > **Regarding Related Work**
> > > > >
> > > > > **Perceiver-AR [1] allows the Perceiver to also perform autoregressive modeling by using causal masking. These recent developments need to be discussed and differences highlighted.**
> > > > >
> > > > > We are happy to cite this work. One can think of characterizing the idea along 3 axis:
> > > > >
> > > > > - multi-stream: Multiple streams with different parameters and operating on "different-kinds of information".
> > > > > - multi-component: Each stream is composed of "multiple" modules.
> > > > > - multi-scale: Multiple streams operate at different time-scales.
> > > > >
> > > > > The proposed architecture is multi-stream, multi-component and multi-scale. Perceiver-AR is multi-component. It iteratively does self-attention (among-tokens), and cross-attention (to the larger context). We have used the available source code provided by the authors, and found this to generally work worse than the normal transformers (we have tried this for LRA benchmark and BabyAI using exactly the same code). In this work, we propose an architecture that has information asymmetry property as an explicit inductive bias. Fast stream has information from local context, and slow stream has information from distant past. Fast stream relies on information from slow stream for relevant contextual information. This "asymmetry" forces the representation to capture relevant contextual information.
> > > > >
> > > > > Across different experiments (LRA, language modelling, visual classification, imitation learning, self-supervised learning), we show the usefulness of the proposed method. Also, for language modelling Perceiver-AR seems to work comparable to Transformer-XL, while we seem to improve the performance over Transformer-XL. It's possible Perceiver-AR works better when used with more capacity (they use 60 layers for some of their experiments).
> > > > >
> > > > >
> > > > > **Improvements made to Slot Attention as well, such as SAVi [2] which allows for the set of slots to maintain state and remain stable over a full video sequence.**
> > > > >
> > > > > Making the slots stable over a full video sequence was actually NOT the contribution of SAVi. This was actually already done in "Objects file and Schemata:Factorizing Declarative and Procedural Knowledge in Dynamical Systems" ICLR'21 [1]. We are happy to cite this work as general application of slot based models for video.
> > > > >
> > > > > [1] Objects file and Schemata:Factorizing Declarative and Procedural Knowledge in Dynamical Systemshttps://arxiv.org/abs/2006.16225
> > > > >
> > > > > We would like to thank the reviewer for taking the time to respond to our rebuttal. We hope that we have addressed all the concerns of the reviewer. We would like to know if there are any more concerns we could address or any more experiments we can run in order to change the reviewer's perception of the paper (and the score).

---

> ### Author Response · Authors · 2022-08-09
> **Addressing reviewer's concerns in initial review**
>
> Dear. Reviewer,
>
> We thank the reviewer for your efforts in giving detailed feedback and for reviewer's time. We thoroughly appreciate it.
>
> To summarize in the initial review, reviewer asked these questions:
>
> (a) **Related work and novelty**: In the initial review, reviewer asked how TLB relates to Perceiver, Slot Attention, DCEM etc. We addressed it.
>
> (b) **Training data and memory**: We have reported these numbers for TLB. Thanks for acknowledging it.
>
> (c) **Insights**: We have added insights as to what different components are relevant for TLB, with comparison to various strong baselines.
>
> (d) **Seeds**: Reviewer mentioned inconsistency between varying number of seeds  (3 v/s 5 v/s 10). We will make sure CIFAR experiments has total 5 seeds (Table 1 and 2, though we did not notice any significant variance).
>
> (e) **Block Transformer**:  After reviewer's suggestions, we ran more experiments comparing to Block Transformer and even reported results keeping EVERYTHING same.
>
> (f) **Introduction**: Following reviewer's suggestion, we modified introduction and it seems the reviewer feel satisfied by the new introduction. We thank the reviewer for reading the new introduction and for their  time.
>
> We have responded to all the concerns in reviewer's initial review (except seeds and table/bar-plot discrepancy, which we assure we will fix for camera ready). We are happy to provide any further clarifications, and happy to incorporate suggestions that would improve the paper.  We feel like these changes addresses all the reviewer's concerns.
>
> In general, more experiments can always be done to test the proposed method, but given the number of different baselines, performance on a wide variety of domains), we feel like we have already shown the breadth and width of the utility of the proposed method.
>
> We would be grateful if the reviewer can provide more feedback and revise their score/rating accordingly.
>
> We appreciate the reviewer's time and feedback.
>
> Thank you.

---

### Official Review · Reviewer_scsa · 2022-07-11

**Rating:** 6
**Confidence:** 4
**Soundness:** 3 good
**Presentation:** 4 excellent
**Contribution:** 3 good

**Summary:**

The authors propose a modification to the transformer architecture called Temporal Latent Bottleneck (TLB). The motivation is to provide a more compressed representation by forcing the model to pass chunks of the input stream through a bottleneck. Whereas a standard transformer may attend to any part of the sequence when encoding the current element, the TLB transformer can only attend to elements in the current chunk; the information beyond this chunk is received only through the bottleneck vector.

The authors maintain that their modified transformer architecture can be applied in any setting where standard transformers were previously used. They provide a series of demonstrations spanning image perception, sequential decision making, and language modeling. In each of these domains, the proposed model outperforms alternative transformer baselines.

**Questions:**

My major suggestion is outlined above. In addition I have a smaller suggestion about the enwiki8 experiment (section 4.3). This experiment is described in a total of 2 sentences with very little detail about the task, the choice of baseline, or the evaluation criteria. There is also little to no reflection provided about the results. I suggest either removing this experiment or expanding it to a complete description.

**Limitations:**

The limitations are adequately addressed.

**Strengths And Weaknesses:**

The writing throughout the paper is clear and technically sound. Moreover the quality of the experiments is high. The originality I will rate as low to medium: although the authors describe a grand motivation for the architectural ideas & principles of their proposed model, the proposal is ultimately only a modification to the existing transformer, and there is little support provided to justify the claims made in their introduction (see specific comments below). Nevertheless, the work has a high potential impact and significance given that transformers have become engrained in almost every area of ML/AI, and the authors have demonstrated the effectiveness of their approach in several different transformer use cases.

My biggest suggestion to improve this paper is the following: I would suggest scaling back the tone and broader framing of the introduction to focus on providing a new transformer architecture that outperforms the default transformer and that is applicable to a number of different settings. At the moment, there’s a major emphasis placed on architectural ideas/principles which are not adequately studied. These include, but are not limited to:
1.  Bottlenecked/compressed representation for improved sample complexity. Throughout the intro the authors talk about learning a more compressed representation, with the motivation being that a more compressed model should have better sample complexity. However, there is very little discussion or exploration of sample complexity in the provided experiments. In the CIFAR experiments, for example, it’s unclear how many training examples are being used, or whether there is a certain sample complexity at which alternative models begin to outperform the proposed. The idea of sample complexity is touched on very briefly in the Baby AI experiment, although “number of timesteps” is not a perfect analogue to sample size (there may be repeated examples over any period of time, and a model with strong sample complexity could have slow convergence for other reasons).
2.  Top-down conditioning. The authors suggest that, by conditioning the chunk encoder F on the latent bottleneck state, the model is required to reason in a “top-down” manner, where information from a higher level of abstraction informs the encoding of low-level raw inputs. They justify this claim with an ablation study (Table 2) where they show that, when they remove the top-down connection in TLB, performance deteriorates. Although I think this ablation is interesting, I don’t think it provides support that the model is reasoning “top-down” as the authors say. Given the constraint of the chunking architecture, the ablated model has no form of information about the past whatsoever, at any level of abstraction. It seems obvious that the model would underperform in this configuration, and I don’t feel that it evidences the use of higher level abstraction.

---

> ### Author Response · Authors · 2022-08-02
> **On framing of the introduction and sample complexity**
>
> We are glad the the reviewer found that the work has high potential impact and its significance. We would like the thank the reviewer for all their suggestions in improving the writing and we will make the appropriate changes. We have addressed some of the concerns raised by the reviewer below:
>
> We would first like to give a general update that we were able to improve the performance of the proposed model on the text classification task from the long range arena benchmark. We now achieve a performance of 82.08% as compared to 67.8% that we reported in the submission. This beats the highly competitive S4 model [3] which achieves 76.02%. The main improvement comes from reducing chunk size to 10 (+6%) and adding local positional embedding within each chunk (+8%).
>
> **On the framing of the introduction**
>
> We have updated the introduction based on the suggestion from the reviewer. We hope that the improved introduction is clearer now. We appreciate the time and feedback of the reviewer.
>
> We have also removed the language modelling experiment as per the reviewers suggestion.
>
> We thank the reviewer for these suggestions. We would be happy to make any other changes that the reviewer suggests.
>
> **On Sample Complexity**
>
> The reviewer has asked for more discussion sample effeciency of the model. Previously, we had experiments showing that proposed Transformer + TLB model converges faster than the baseline Transformer on the Baby AI benchmark. The reviewer says that number of training steps is not a good indicator of sample complexity. We agree with the reviewer and therefore we further evaluate the sample complexity of the model in two more settings.
>
> - **Sample Efficiency in ListOps**: We evaluate the sample efficiency of the proposed Transformer + TLB model and the baseline Transformer LS model on the ListOps dataset. For this, we train both the models for one pass over the entire ListOps dataset i.e. no example is seen more than once by the model. We plot the convergence curves against the number of samples in Appendix Figure 7 (a). We can see that the Transformer + TLB model is much more sample efficient than the Transformer LS baseline converging in almost half as many examples as the baselines.
> - **Sample Efficiency in Copying Task**: Additionally, we also evaluate the proposed Transformer + TLB model on the copying task used in [1]. In this task the model is first given a sequence of 10 digits as input and it is supposed to output the same sequence after an interval of blank inputs. We describe the task in detail in Section 3.4 of the paper. The length of the blank inputs in the middle can be as high as needed resulting in very long sequence lengths. For a model to do well, it needs to remember the original sequence across long time-scales. We compare the proposed Transformer + TLB model to the Feedback Transformer [2] which has been shown to work well on this task. We compare both the models for varying sequences lengths from 100 to 600 as shown in Figure 5 in the main paper. We see that for higher sequence lengths the performance of the Feedback Transformer starts dropping while the proposed model still achieves perfect accuracy. We also evaluate the sample efficiency of both the models on the copying task in similar manner as for the ListOps task. Both the proposed model and the baseline Feedback Transformer see only unique examples during training. We calculate the number of samples required for both the models to achieve perfect accuracy.
>     | Sequence Length | Feedback Transformer | Transformer + TLB |
>     | --------------- | -------------------- | ----------------- |
>     | 100             | 11800                | 6200              |
>     |200 | 16600 | 9100 |
>     | 300 | 35100 | 12700 |
>     |400 | NA | 14600 |
>     | 500 | NA | 13600 |
>     |600 | NA | 19300 |
>
>     We can see that in all cases Transformer + TLB needs much lesser samples than the baseline to converge to perfect accuracy. NA in the above table indicates that the model does not converge to perfect accuracy.
>
> [1] Long Short-Term Memory Hochreiter et al 1997
>
> [2] Addressing Some Limitations of Transformers with Feedback Memory Fan et al 2021
>
> [3] Efficiently Modeling Long Sequences with Structured State Spaces Gu et al 2021

---

> > ### Author Response · Authors · 2022-08-02
> > **On importance of Top-Down Commuinication**
> >
> > **On the importance of Top-Down information**
> >
> > The reviewer says that there is no ablation where we justify that top-down information is being used. The current ablation that we have in which we show a performance drop on removing top-down information is expected since the resulting model has no access to the past. To address this, we conduct an ablation to show that the TLB is not only important because it conveys information about the past but also because it conveys high-level information to the lower levels through top-down feedback. We design a baseline which is similar to our proposed Transformer + TLB model but it has a different TLB module after each layer as opposed to having only one TLB module after the last layer. Thus, each layer L of the perceptual module reads and writes information from the TLB module specific to layer L. Therefore, in this case the TLB module only provides information about the past, it does not contain any top-down information since the TLB module at layer L does not communicate with the lower layers of the perceptual module. We evaluate this baseline on CIFAR10 image classification, IMDB text classification, and ListOps. The results are as follows:
> >
> > - CIFAR10 Image Classification - ViT + TLB: 94.79%, ViT + TLB (No Top-Down): 93.75%
> > - ListOps: Transformer + TLB: 38.2%; Transformer + TLB (No Top-Down): 37.57%
> > - IMDB Text Classification - Transformer + TLB: 82.08%; Transformer + TLB (No Top-Down): 80.68%
> >
> > This shows that the TLBs ability to provide top-down information along with information from the past is important for the model to perform well. Only providing summarized information from the past is not enough.
> >
> > We hope that the rebuttal addresses all the concerns and questions raised by the reviewer. We would be happy to address any more concerns that the reviewer may have. We would also be happy to perform any more experiments that the reviewer may want us to perform.

---

> ### Author Response · Authors · 2022-08-08
> **Follow up on rebuttal**
>
> Dear. Reviewer,
>
> Thanks again for taking the time to review our paper and providing feedback. We have updated the introduction of the paper, as well as added detailed replies to all of the reviewer's comments (after conducting more experiments). We note that two concerns of the reviewers were (a) the importance of top-down information, and (b) the sample complexity of the proposed method.
>
> Through our experiments performed in the rebuttal period, we have shown that the TLB is not only important since it provides information from the past but also because it provides top-down information (page 8 last paragraph and page 5 last paragraph). We have also shown that TLB is more sample efficient on ListOps and the newly performed copying task. Here, we have calculated sample complexity as the number of unique samples as the reviewer requested (Appendix Figure 7(a) and Table 5).
>
> We felt that the reviewer's review was very helpful and instructive, so it would be very helpful to get feedback on our changes (and also update the rating if applicable). We would also be grateful if the reviewer could let us know if there are any other concerns that we can address further, which would improve our work, and may assist the reviewer in revising their opinion.
>
> Thanks for your help and time. :-)

---

> ### Comment · Reviewer_scsa · 2022-08-09
> **Response to author rebuttal**
>
> Thank you for answering my questions and addressing my concerns, and apologies for the delayed reply. The new introduction is indeed much improved. I think the paper is good enough to be accepted, and after a careful review of the author reply I have decided to keep my current score & recommendation.
>
> The new ablation study is more convincing than the original, although I am not entirely clear about the architecture of the new baseline model (no top-down condn) and I feel that it would benefit from a visual.
>
> The additional experiments studying sample complexity are a start; however, these experiments were added only to the appendix, and sample complexity still feels like an afterthought. I stand by my assessment that it should not be a major focus of the introduction/motivation.

---

> > ### Author Response · Authors · 2022-08-09
> > **Thank you**
> >
> > Dear. Reviewer,
> >
> > We thank the reviewer for taking time, and reading our rebuttal.
> >
> > **The new introduction is indeed much improved**
> >
> > We are enthused that the reviewer likes the new introduction.
> >
> > **I think the paper is good enough to be accepted**
> >
> > We are happy to hear. Thank you.
> >
> > **I feel that it would benefit from a visual**
> >
> > We will update the paper with a visual. Thank you for your suggestion.
> >
> > **however, these experiments were added only to the appendix**
> >
> > We will incorporate sample complexity curves in the main paper.
> >
> > Please don't hesitate to let us know if there are any other concerns which we can address.
> >
> > Thank you for your time, and help.

---

### Official Review · Reviewer_p6Zz · 2022-07-15

**Rating:** 7
**Confidence:** 3
**Soundness:** 3 good
**Presentation:** 4 excellent
**Contribution:** 3 good

**Summary:**

This paper proposes Temporal Latent Bottleneck (TLB) to help transformer-based models benefit from the compression of sequential inputs while not having to (significantly) forego their expressivity. This is done by dividing the computation stream into a slow stream that is recurrent in nature, and a fast stream that allows for attention mechanism between tokens. One of the salient abilities of the proposed architecture is the mutual interaction between both the streams, through a top-down connection between the recurrent and transformer modules (allowing for cross attention), and a bottom-up connection (allowing the recurrent module to take as inputs the final layer of transformer module). Experiments are performed across Vision, Language, and RL domains to show that the improvements brought by the proposed architecture are consistent across domains.

**Questions:**

Mentioned above

**Limitations:**

Memory and Time complexity should be talked about. While the computation complexity does decrease, the time complexity is expected to rise as each individual attention block waits for the previous block, and the corresponding RNN module to complete and provide input for the cross attention layer. I would also like to see the difference in training and inference time. Since GPU parallelization is used pervasively, reporting computation complexity is an unfair metric.

**Strengths And Weaknesses:**

## Strengths
1. The paper does a good job of explaining details of the newly proposed architecture.
2. Comparisons with baseline methods over multiple domains are convincing and the performance improvements are consistent.

## Weakness
1. The underlying hypothesis to motivate the introduction of a recurrent stream was that it may benefit generalization since it may contain lesser irrelevant details. I find (a) this hypothesis unconvincing, and (b) no evidence in the paper to support the same.
2. The paper motivates the use of a transformer architecture by arguing how past work has found recurrent architectures to be incapable of remembering long-range dependencies. In light of the same, it would make a lot more sense if the recurrent stream of the input happened in chunks of K-sized tokens, and the transformer part acted upon the outputs of each layer. This would allow the recurrent stream to do what it is good at (compressing not-so-long range dependencies), and the attention part to attend to all pairs over compressed tokens from anywhere in the input. The current architecture choice appears misplaced, especially based on the motivation that the authors use to position the proposed architecture.
3. One of the claims of the work is about the ability of the architecture to perform well in resource-constrained scenarios. I would have liked to see an analysis of how the model fairs as more pertaining data becomes available (like in typical language models). Do the advantages of such an architecture still hold up? Will such an architecture be of relevance to the deep learning community at large? Even in case of resource-constrained settings, I find an important analysis missing about the performance change with sequence length. Since the attention module is limited by its attention window to K chunks, this may have an unwanted impact as the sequence length changes. In this regard, text classification on datasets of long sentence length (like IMDb) and short sentence lengths (like SST) would be an interesting point of comparison.

## Other Comments
1. The use of vspace is very evident. Please take it into consideration.
2. In Figure 3, it appears that the baseline method has not yet converged. Did you train for longer?
3. In Figure 3, should the accuracy for Multitask be comparable to a single task?
4. The computation cost seems incorrect. There would be an additional term for (T/K) for each of the K-sized modules (line 90). This would of course still be less than O(t^2)
5. Can you use TLB with Swin Transformer?
6. Why are you not using a bidirectional recurrent layer. Bidirectional RNNs and LSTMs are known to outperform traditional ones. It appears that except for the sequential tasks (4.2), using a bidirectional recurrent layer should improve performance.
7. What is the time taken for training and inference as opposed to baseline methods?

---

> ### Author Response · Authors · 2022-08-02
> **Analysis of the information in recurrent stream and importance of chosen components to implement both the streams**
>
> We thank the reviewer for their constructive feedback and questions. We have updated the paper with the feedback in mind. We have addressed some of the concerns and comments below:
>
>
> We would first like to give a general update that we were able to improve the performance of the proposed model on the text classification task from the long range arena benchmark. We now achieve a performance of 82.08% as compared to 67.8% that we reported in the submission. This beats the highly competitive S4 model [3] which achieves 76.02%. The main improvement comes from reducing chunk size to 10 (+6%) and adding local positional embedding within each chunk (+8%).
>
> **On the hypothesis that reccurrent stream captures less irrelevant information**
>
> The reviewer says that there is no evidence in the paper that shows that the recurrent stream captures irrelevant information. To show evidence of this claim we perform two experiments:
>
> - First, for the image classification experiment, we visualize the patches that the TLB pays most attention to. To do this we first store the attention scores that the TLB pays to each patch in the image when the perceptual module is writing information to it (i.e. attention scores computed in the last step of Algorithm 1 in the Appendix). We only keep those patches in the image that fall in the top 30% of the attention scores masking everything else. We present the result of this visualization experiment in Figure 2 in the main paper. We can see that the TLB pays most attention to foreground of the image which shows that the TLB focuses on the relevant information (i.e, foreground) in the image learning to ignore irrelevant information.
> - Second, we evaluate the proposed Transformer + TLB model on the copying task used in [1]. In this task the model is first given a sequence of 10 digits as input and it is supposed to output the same sequence after an interval of blank inputs. We describe the task in detail in Section 3.4 of the paper. The length of the blank inputs in the middle can be increased (i.e., distractor information) resulting in very long sequence lengths. For a model to do well, it needs to remember the original sequence across long time-scales. We compare the proposed Transformer + TLB model to the Feedback Transformer [2] which has been shown to work well on this task. We compare both the models for varying sequences lengths from 100 to 600 as shown in Figure 5 in the main paper. We see that for higher sequence lengths the performance of the Feedback Transformer starts dropping while the proposed model still achieves perfect accuracy. This shows that the TLB retains the original sequence across long time scales without overwriting any of its information in the presence of blank inputs which shows that it captures and retains the most important information for solving the task.
>
> **On flipping the Recurrent and Transformer Streams**
>
> The reviewer says that having the low level as the recurrent stream and high-level as the transformer would allow the recurrent stream to do what it is good at (i.e. compressing short-term information). While this proposal makes sense, this is not in line with our motivation. Our motivations behind the proposed architecture are as follows:
>
> - We want the low-level fast stream to be a high capacity stream which has high representation power strong perceptual abilities. Transformers have demonstrated strong performance in many perceptual and have shown to have high representation capacity this makes them the ideal choice for the low-level. On the other hand, we want the high-level slow stream to be of limited capacity so that it captures the most important information in the input. Recurrent neural networks are ideal for this since they have limited capacity due to high rate of compression. Also, we want a model that can forget information from the distant past that is not relevant in the current context. If we use a Transformer, it would capture all pairwise including those that include chunks from the distant past thus resulting in no summarization/compression of information.
> - Secondly, one advantage of the proposed model is that the computational complexity of the attention is much less than quadractic and remains constant as we increase the number of chunks. If we introduce a Transformer in the higher level, the computational complexity will increase quadratically in the length chunks.
>
> [1] Long Short-Term Memory Hochreiter et al 1997
>
> [2] Addressing Some Limitations of Transformers with Feedback Memory Fan et al 2021
>
> [3] Efficiently Modeling Long Sequences with Structured State Spaces Gu et al 2021

---

> > ### Author Response · Authors · 2022-08-02
> > **Analysis of the model performance with increasing sequence lengths and responding to reviewer questions**
> >
> > **On sequence length analysis and performance under more pretraining data**
> >
> > The reviewer asks how would the model scale as more pretraining data becomes available in language modeling. The reviewer also asks for an analysis of the performance as the sequence length changes. We are unable to perform an analysis of the performance as a result of more pretraining data in language modeling due to a lack of compute resources and limited time. However, we do perform an analysis of the performance of the model with changing sequence lengths for the copying task as described above. In Figure 5 of the main paper, we can see that the performance of the proposed Transformer + TLB model stays the same as sequence length increases but that of the Feedback Transformer drops. We would also like to point that the byte-level text classification and the ListOps experiment from the long range arena benchmark have extremely long sequence lengths from 1000-4000. The byte-level text classification task from long range arena uses the IMDB reviews dataset.
> >
> > **Reviewer Questions**
> >
> >
> > - *In Figure 3, should the accuracy for Multitask be comparable to a single task?*
> >
> >   For the multitask plot, we report the average accuracy across all tasks. We find that the individual accuracies for all the tasks are in the range from 70%-90% resulting in an average multitask accuracy of ~80%. It is possible that the multitask accuracy is same as single task since language interface and grid structure of all tasks is shared.
> > - *The computation cost seems incorrect. There would be an additional term for (T/K) for each of the K-sized modules (line 90). This would of course still be less than O(t^2)*
> >
> >   We thank the reviewer for pointing this mistake, we have updated the main paper accordingly.
> > - *Can you use TLB with Swin Transformer?*
> >
> >   Yes, in principal the perceptual module can be any architecture of choice. Therefore, the Transformer that processes the chunk can be replaced by a swin transformer.
> > - *Why are you not using a bidirectional recurrent layer. Bidirectional RNNs and LSTMs are known to outperform traditional ones. It appears that except for the sequential tasks (4.2), using a bidirectional recurrent layer should improve performance.*
> >
> >   We cannot use a bidirectional RNN in the high-level recurrent stream since it is inherently autoregressive in nature. Bidirectional RNNs can only be used when the entire sequence is already available. In our case, the input to the RNN is the output of the perceptual module at each chunk. The output of the perceptual module is generated auto-regressively one by one since it has to incorporate the top-down feedback from the TLB at each step.
> > - *What is the time taken for training and inference as opposed to baseline methods?*
> >
> >     We evaluate the memory and wall-clock times for the proposed model w.r.t to the baseline Transformer model on the text classification task from the LRA benchmark (sequence length of around ~4k). We present the results in the following table. In the following table x indicates the value of the corresponding metric for a vanilla transformer.
> >
> >     | Chunk Size | 1000 | 100 | 40 | 20 | 10 |
> >     | ---------- | ---- | --- | -- | -- | -- |
> >     | Inference Speed |  3.5x | 3.6x | 3.3x | 2.2x | 1.2x |
> >     |    Inference Memory |  0.09x | 0.08x | 0.12x | 0.08x | 0.1x |
> >     |    Training Speed |  4.4x | 4.4x | 2.2x | 1.4x | 0.7x |
> >     |    Training Memory |  0.14x | 0.08x | 0.49x | 0.40x | 0.42x |
> >
> >     Therefore, a Transformer + TLB model with chunk size 20 has 2.2x faster inference than a Transformer, uses 0.08x inference memory of a Tranformer, trains 1.4x times faster than a Transformer, and uses 0.40x memory of a Transformer during training. We refer the reviewer to Appendix section 7.3 (last paragraph) for more details on this. We can see that except for chunk length 10, for all other cases the proposed model trains faster than the vanilla transformer baseline.
> >
> > -  *In Figure 3, it appears that the baseline method has not yet converged. Did you train for longer?*
> >
> >      We have trained the baseline model and the Transformer + TLB model for 100k steps as compared to 50k and updated the Figure (now it is Figure 4 in the main paper) in the paper. We find that both the models seem to be converging to the same value.
> >
> > We hope that the rebuttal addresses all the concerns and questions raised by the reviewer. We would be happy to address any more concerns that the reviewer may have. We would also be happy to perform any more experiments that the reviewer may want.

---

> > > ### Comment · Reviewer_p6Zz · 2022-08-05
> > > **Follow Up**
> > >
> > > Thank you for your detailed response. I am happy to maintain my assessment.

---

> > > > ### Author Response · Authors · 2022-08-06
> > > > **Thank you**
> > > >
> > > > We thank the reviewer for taking time and reading our response.

---

### Author Response · Authors · 2022-08-02
**Summary of new experimental results ran during the rebuttal phase**

We thank the reviewer for their very useful comments and suggestions for experiments, which we think will make our papers much stronger. We are able to finish most of these. To summarize, reviewers asked for the following experiments:

We would first like to give a general update that we were able to improve the performance of the proposed model on the text classification task from the long range arena benchmark. We now achieve a performance of 82.08% as compared to 67.8% that we reported in the submission. This beats the highly competitive S4 model [3] which achieves 76.02%. The main improvement comes from reducing chunk size to 10 (+6%) and adding local positional embedding within each chunk (+8%).

We have conducted more experiments analysing various components of the proposed model -

- **Effect of top-down attention**: To probe the effect of top-down attention, we conduct an experiment where we compare to a baseline with no top-down communication. We present a comparison to this baseline for CIFAR10 Image Classification (Quantitative Analysis in Section 3.1 in main paper) and  ListOps (section 3.3 in main paper). We show that our proposed model performs well against this baseline.
- **TLB pays attention to relevant information**: We visualize the information that gets written to the TLB in the CIFAR10 Image Classification Experiments. We find that the TLB learns to almost perfectly attend to the foreground showing that it learns to focus on the important information.
- **Comparison to Feedback transformers**: We add a new experiment on the copying task where we compare the proposed model to the Feedback Transformer model. We analyse how the performance of both the models change with increasing sequence lengths. We find that while the performance of Feedback Transformer drops at high sequence lengths, the performance of our model remains high.
- **Sample efficiency of the proposed model**: We present more results showing that our model is more sample effecient than various baselines. For ListOps, we show in Appendix Figure 7 (a) that the proposed model converges in almost half the number of unique samples than the baseline. For the copying task also we show in Table 5 that the proposed model converges in much lesser number of samples than Feedback Transformer.
- **TLB retains information across long time-scales**: To show that the model can retain information over long time scales, we compare the model to a baseline where each chunk attends to the past 4 chunks. We conduct this experiment on the ListOps task and find that this baseline achieves a performance of 32.10% while we achieve a performance of 38.2%.
- **Effect of Chunk Size and Number of State vectors**: We also conduct experiments to analyse the effect of the chunk size and number of state vectors on the performance of the model in the ListOps task. We present this experiment in Appendix section 7.3. We find that for both there is an optimal number of the hyperparameter above or below which the performance drops.
- **Consequences of Clock based Chunking**: We analyse the consequences of static chunking (i.e., fixed chunking). To do this we conduct an experiment on the text classification benchmark where we introduce spaces into the sequences such that each word is divided across two chunks. Therefore, each chunk sees the second half of the previous word and first half of the next word. We find that the performance of the model drops from 82.08% to 81.29% which is not a significant but this still highlights the limitation of static chunking which future work should address.

**Memory and Time Requirements for the Proposed Model**

We compare the time and memory requirements for the proposed model to those of a Transformer on the byte-level text classification task which has extremely long sequence lengths of upto 4k. We obtain the following results:

| Chunk Size | 1000 | 100 | 40 | 20 | 10 |
| ---------- | ---- | --- | -- | -- | -- |
| Inference Speed |  3.5x | 3.6x | 3.3x | 2.2x | 1.2x |
|    Inference Memory |  0.09x | 0.08x | 0.12x | 0.08x | 0.1x |
|    Training Speed |  4.4x | 4.4x | 2.2x | 1.4x | 0.7x |
|    Training Memory |  0.14x | 0.08x | 0.49x | 0.40x | 0.42x |

Here x refers to the value of corresponding metric for the vanilla transformer. We find that the proposed model is less memory intensive the transformer and is faster in training than the Transformer except when chunks lengths are very low.

We have updated the paper with all these experimental results and have also incorporated the various writing suggestion made by each reviewer. We hope that these experiments will address any remaining concerns that the reviewers may have.

---

### Author Response · Authors · 2022-08-09
**Summarizing the reviews and our rebuttal**

We thank all the reviewers for their time and feedback.  We are enthused by the positive feedback from all the reviewers: "paper does a good job of explaining details, Comparisons with baseline methods are convincing, performance improvements are consistent" (Reviewer p6Zz), "writing  is clear, quality of the experiments is high, paper is good enough to be accepted" (Reviewer scsa), "high-level idea is interesting, shows very good empirical gains on Atari in the offline setting, like the high-level idea, appreciate the efforts of the authors to empirically test on a wide variety of domains" (Reviewer EwzR).

We briefly summarize the main concerns and explain how we addressed them:

List of main concerns -
1. **Introduction Concern**: Introduction needs to be improved.
2. **Analysis Concern**: More analysis of the model required
3. **Baseline Concerns**: The model has not been tested against competitive baselines that have some of the same components as the proposed TLB model.

**Reviewer P6Zz** (original rating: 7; new rating: 7)

| Concern | Summary of rebuttal | Location where concern addressed |
| ------- | ---- | ---- |
| No evidence that recurrent stream contains less irrelevant information | Demonstrated through visualizations and experiments that this is indeed the case. | 1. Visualization of information written into the TLB (Figure 2 in main paper) <br /> 2. Outperforming feedback Transformer on copying task where ignoring irrelevant information is critical (Section 3.3 last 2 para in main paper) <br /> |
| Recurrent stream and transformer stream should be flipped to align more with the paper's motivation | This would lead to a higher computational complexity than the TLB model | Address in detail in https://openreview.net/forum?id=mq-8p5pUnEX&noteId=\_atbOqx5LP |
| Need analysis of effect of increase of sequence length on model performance | Experimentally showed that proposed TLB model is more robust than baselines on increasing sequence length | Addressed in Figure 5 in main paper |


**Reviewer scsa** (original rating: 6; new rating: 6)
| Concern | Summary of rebuttal | Location where concern addressed |
| ------- | ---- | ---- |
| Introduction Concern | Improved the introduction. | Section 1 main paper |
| Sample effeciency of the model. | Conducted more experiments to show that the TLB is more sample efficient. | Table 5 main paper and Appendix Figure 8(a). |
| Importance of Top-Down Attention not shown | Conducted an experiment to show the importance of top-down attention. | Addressed in https://openreview.net/forum?id=mq-8p5pUnEX&noteId=JHLMfv08NhX |

The reviewer was satisfied by the rebuttal and makes the remark that **"I think the paper is good enough to be accepted"**.

**Reviewer CN5N** (original rating: 4; new rating: 5)
| Concern | Summary of rebuttal | Location where concern addressed |
| ------- | ---- | ---- |
| Difference between our method and different baselines | We have explained the difference | Addressed in https://openreview.net/forum?id=mq-8p5pUnEX&noteId=uLMgAfUJbYN |
| Baseline Concern | Baselines considered are extremely competitive and achieve state-of-the-art results in some cases. | Addressed in  https://openreview.net/forum?id=mq-8p5pUnEX&noteId=-ZDUyQRYak8 |
| Comparison against Feedback Transformer |Proposed model outperforms Feedback Transformer on the copying task. | Section 3.3 last 2 para in main paper. |
| Analysis Concern | Ran experiments analysing the behavior of our model. | We summarize these experiments in https://openreview.net/forum?id=mq-8p5pUnEX&noteId=9Y9js5j7H749 |

The reviewer further raised the concern that RIMs can capture multiscale information similar to TLB. We have addressed this in https://openreview.net/forum?id=mq-8p5pUnEX&noteId=prlZsseUgUW.

---

> ### Author Response · Authors · 2022-08-09
> **Summarizing the reviews and our rebuttal  (contd)**
>
> **Reviewer EwzR** (original rating: 4; new rating: 4)
>
> The reviewer raised detailed concerns in the initial review as well as in the discussion phase. We thank the reviewer for the detailed review. We feel that we have addressed _every_ concern raised by the reviewer.
>
> Detailed Concerns 1 (Before the rebuttal phase)
> | Concern | Summary of rebuttal | Location where concern addressed |
> | ------- | ---- | ---- |
> | Difference between our method and different baselines | We have explained the difference | Addressed in https://openreview.net/forum?id=mq-8p5pUnEX&noteId=\_906nMCZKbW |
> | Writing introduction and various other writing issues | We have updated introduction and tried to fix all writing and presentation issues | Addressed Section 1 in main paper |
> | Training time, Memory and  sample effeciency | We have addressed this. | 1. Gains in training time and memory shown in Appendix Table 8. <br /> 2. Gains in sample complexity shown in Table 5 in main paper and Appendix Figure 8(a) | Analysis Concern | We have run many experiments analysing the behavior of our model | We summarize these experiments in https://openreview.net/forum?id=mq-8p5pUnEX&noteId=9Y9js5j7H749 |
> | Baseline has not converged yet in Figure 4 | We have run for 100k steps. Both baseline and TLB have converged now. | Figure 4 (left) in main paper |
> | Seeds used not consistent | For CIFAR experiments (Table 2, 3) we will run for 5 seeds (as compared to 3).  |  |
> | Comparison to block-recurrent transformer | Block recurrent Transformer is parallel work. While they focus mainly on NLP tasks, we focus on a broader variety of tasks. We conducted experiments comparing the two methods. | Addressed in https://openreview.net/forum?id=mq-8p5pUnEX&noteId=jBDD1We37GxO |
>
> The reviewer appreciated the experiments we performed for the rebuttal and raised more concerns about the paper and the rebuttal. We have detailed these concerns below.
>
> Detailed Concerns 2 (During discussion phase)
> | Concern | Summary of rebuttal | Location where concern addressed |
> | ------- | ---- | ---- |
> | Computational complexity analysis is in appendix | Will shift this to main paper. | |
> | More analysis on Figure 2 requested | We have added. | Addressed in Appendix Figure 6 |
> | No standard deviations reported in Tables 1, 2. | We have addressed this | https://openreview.net/forum?id=mq-8p5pUnEX&noteId=AOLMXeSg2E1 |
> | Algorithm 1 should be in main paper | We will move it after the rebuttal | |
> | Difference of our method from Perceiver AR and SaVI | We have addressed this |  https://openreview.net/forum?id=mq-8p5pUnEX&noteId=OVPApKzVMuq |
> | Baseline Concern | Baselines considered are extremely competitive and achieve state-of-the-art results in some cases. | Addressed in https://openreview.net/forum?id=mq-8p5pUnEX&noteId=TD61l-QX1- |
> | Is comparison to block-recurrent transformer like-for-like | yes | Addressed in  https://openreview.net/forum?id=mq-8p5pUnEX&noteId=AOLMXeSg2E1 |

---

### Meta-Review · Area_Chair_ucNZ · 2022-08-29

**Recommendation:** Accept
**Confidence:** Less certain

**Metareview:**

This paper proposes a new architecture consisting of both a recurrent and transformer-based component. The reviewers found the performance of the model to be impressive across multiple domains, but were less convinced by some of the quasitechnical claims in the storytelling (even Reviewer p6Zz, who champions the paper expressed this concern). I encourage the authors to be a but more disciplined about these claims in the camera ready version if accepted. Overall the authors engaged admirably in the discussion period providing a strong effort to improve the paper and digestible summaries of the original complaints and corresponding changes in the improved draft.

**Award:**

No

---

### Decision · Program_Chairs · 2022-09-14

Accept